# Don't forget the nullspace! Nullspace occupancy as a mechanism for out of distribution failure

**Daksh Idnani**
Meta (FAIR)

**Vivek Madan**
Amazon

**Naman Goyal**
Meta (FAIR)

**David J. Schwab**
Meta (CTRL)

**Ramakrishna Vedantam**
Meta (FAIR)

## Abstract

Out of distribution (OoD) generalization has received considerable interest in recent years. In this work, we identify a particular failure mode of OoD generalization for discriminative classifiers that is based on test data (from a new domain) lying in the nullspace of features learnt from source data. We demonstrate the existence of this failure mode across multiple networks trained across RotatedMNIST, PACS, TerraIncognita, DomainNet and ImageNet-R datasets. We then study different choices for characterizing the feature space and show that projecting intermediate representations onto the span of directions that obtain maximum training accuracy provides consistent improvements in OoD performance. Finally, we show that such nullspace behavior also provides an insight into neural networks trained on poisoned data. We hope our work galvanizes interest in the relationship between the nullspace occupancy failure mode and generalization.

## 1 Introduction

Neural networks often succeed in learning rich function approximators that generalize remarkably well to the distribution they are trained on, but are often brittle when exposed to inputs that come from a different distribution (Gulrajani & Lopez-Paz, 2020). With rapid adoption of neural networks to various safety critical applications such as autonomous driving, healthcare etc. more attention is being paid to the question of robustness under domain shift (Alcorn et al., 2018; Dai & Van Gool, 2018; AlBadawy et al., 2018).

Recent findings from Huh et al. (2021) hint that overparameterized, deep neural networks are biased to learn functions with (approximate) low-rank covariance structure and posit that this might be related to the phenomenon of implicit regularization (Galanti & Poggio, 2022) that has been used to explain in-distribution generalization of deep networks.

*How might such low-rank structure relate to out-of-distribution generalization?* As a simple thought experiment, consider a setting where training data $\mathcal{D}_{train}$ is embedded in a three dimensional space $(\mathbf{v}_1, \mathbf{v}_2, \mathbf{v}_3)$ that exhibits variance only along the first two dimensions (fig. 1 (left)) (with $\mathbf{v}_3 = 1$) [1]. Let us train a neural network $f_\theta$ on this data using a loss functional $\mathcal{L}(f, \mathcal{D}_{train})$. Since $\mathbf{v}_3$ does not contribute to any reduction in training error, standard empirical risk minimization (ERM) (Vapnik, 1999) training need not differentiate between functions $f$ which handle $\mathbf{v}_3$ in different ways.

Now consider an out-of-distribution (OoD) dataset which has the same structure as the original dataset along $\mathbf{v}_1$ and $\mathbf{v}_2$, but the value of $\mathbf{v}_3$ now has a different value, e.g. 10. In this case, one would incur an error (fig. 1, right) if one learns a function $f$ where $f(\cdot, \cdot, \mathbf{v}_3 = 1) \neq f(\cdot, \cdot, \mathbf{v}_3 = 10)$. Thus, the low-rank simplicity bias, while beneficial for IID generalization (Huh et al., 2021) can potentially cause issues for OoD generalization. In cases where removing the "additional" features observed at test time improves performance (such as in fig. 1) we say that the network incurs *nullspace error* and call the failure mode as "nullspace occupancy".

---

[1]This is a special case of Huh et al. (2021) where the third eigenvalue is 0, instead of being very small

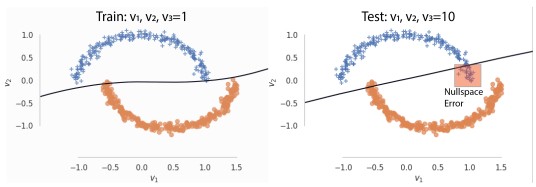

Figure 1: Illustration of nullspace failure. **Left:** Training data with variation in $\mathbf{v}_1$, $\mathbf{v}_2$ but no variation in $\mathbf{v}_3 = 1$. *Black* line: Decision boundary learnt by a 3 hidden layer MLP function $f$ with inputs $(\mathbf{v}_1, \mathbf{v}_2, \mathbf{v}_3)$ visualized with $\mathbf{v}_3 = 1$. **Right:** Decision boundary of the same classifier evaluated on the plane $\mathbf{v}_3 = 10$. $f$ is sensitive to $\mathbf{v}_3$, causing nullspace error (red box) on test data.

When diagnosing nullspace occupancy related failure, it is important to properly choose the representational basis. Our key technical contribution is combining notions of variation in training data with utility for the downstream network $f$ in order to identify the most important directions for projection. We formalize this as an optimization problem that we solve via projected gradient descent onto orthogonal matrices (Kiani et al., 2022).

Experimentally, we demonstrate the existence of nullspace occupancy related performance degradations across different architectures and datasets evaluated on the DomainBed benchmark (table 2). This empirically establishes that nullspace occupancy is an issue for neural networks in OoD settings and suggests further performance improvements by mitigating it. We take first steps towards showing this in practice in the leave-one-out-validation setting from Gulrajani & Lopez-Paz (2020) to improve OoD performance on DomainBed.

Overall, our contributions are as follows:

- We identify a nullspace occupancy based failure mode for OoD generalization
- We demonstrate that this failure mode exists for models trained using ERM on DomainBed
- We observe that selecting a few projecting directions with high training accuracy yields the maximum potential improvements using this approach
- Interestingly, we also find that in a data poisoning setting (Huang et al., 2020), the network exploits this nullspace occupancy phenomenon to learn a poorly generalizing classifier (section 4)

## 2 METHODS

### 2.1 FEATURE PROJECTION MECHANICS

We work in the standard multi-class classification setting with inputs $\mathbf{x}$ and labels $y$. Let $\mathbf{z} = g(\mathbf{x}) \in \mathbb{R}^K$ be the features extracted in an intermediate layer of the network $l$ using an encoder $g$, and $\mu = \frac{1}{N}\sum_{i=1}^{N} \mathbf{z}_i$ be the mean of the features $\mathbf{z}_i$ extracted on a training dataset with $N$ datapoints. Let $f$ be the classification function that maps a feature $\mathbf{z}$ to the logits. Finally, let $V \in \mathbb{R}^{K \times K} = [\mathbf{v}_1, \cdots, \mathbf{v}_K]$ be an orthonormal matrix, and a rank $m$ subspace of $V$ is $V_m \in \mathbb{R}^{K \times m} = [\mathbf{v}_1, \cdots, \mathbf{v}_m]$.

Given a test datapoint $\hat{\mathbf{x}}$, with a corresponding extracted feature $\hat{\mathbf{z}}$, one can project the datapoint onto a basis $V_m$ of rank $m$ as follows:

$$\hat{\mathbf{z}}^m = V_m V_m^T (\hat{\mathbf{z}} - \mu) + \mu \qquad (1)$$

For a convolutional network, we perform projection only along the channel dimension of the featurization. That is, for an intermediate representation with $n_c$ channels and spatial size $H \times W$, we consider a set of features $V_m$ of dimensionality $n_c$. For each spatial location $(i, j) \in [H] \times [W]$, we project a vector $\mathbf{z}_{ij}$ as follows:

$$\hat{\mathbf{z}}_{ij}^m = V_m V_m^T (\hat{\mathbf{z}}_{ij} - \mu_{ij}) + \mu_{ij} \qquad (2)$$

Figure 2: Our methods add projection layers (orange) to an existing pretrained network with an encoder $g_\theta$ (till layer $l$) and downstream classifier $f$.

### 2.2 WHAT IS THE RIGHT CHOICE OF BASIS $V$?

We aim to find a basis $V$ that leads to the highest decrease in the training loss $\mathcal{L}_{train}$ as we increase the rank $m$. This ensures that we cover the most important directions for the loss functional $\mathcal{L}(f, \mathcal{D}_{train})$. Intuitively, this incorporates both notions of training sensitivity as well as feature spread, since directions with a lot of spread of training data and where the function has high sensitivity would also decrease the training loss.

We propose to directly learn the projection matrix that identifies the most parsimonious $V$ based on the training set. The key idea is simple: we initialize the basis $V$ with the principal component basis computed over the training set, *i.e.* $V = V_{pc}$, then denoting $\mu$ as the mean of the features over the training dataset (as above) we solve the following optimization problem, using a projective approach for optimizing unitary matrices (Kiani et al., 2022):

$$\min_V \sum_{m=1}^{K} \sum_{i=1}^{|\mathcal{D}_{train}|} \mathcal{L}_{train}(f(V_m \cdot V_m^T \cdot (\mathbf{z}_i - \mu) + \mu), y_i) \tag{3}$$

subject to the unitary constraint $V^T \cdot V = \mathbf{I}$. Here, the training loss $\mathcal{L}_{train}$ is cross entropy. For a convolutional network, we perform analogous projections along the channel dimension of the featurization. If we consider the training accuracy versus number of components $m$, optimizing this equation will yield the basis which maximizes area under the curve. As a result, for any given target accuracy, we find the smallest value of $m$ that achieves that level of training accuracy, thus learning the most parsimonious basis.

## 2.3 BASELINE CHOICES OF BASES $V$

We consider ablations on the design of the basis $V$ that capture different intuitive notions of what one considers to be important when choosing the features to project onto (see table 1). We explain each in more detail below:

**Principal Components Analysis:** Given the feature matrix $Z \in \mathbf{R}^{N \times K}$ computed over the training data $\mathcal{D}_{train}$, one can compute the principal components analysis (PCA) (Murphy, 2002) of $Z$ and use it as the basis $V$. Intuitively, this captures the directions in the feature space which have the most variance in the training data, but might not capture which directions are used by the downstream function $f$ ( table 1). As we validate in section 4, considering both is important for diagnosing nullspace occupancy related failures.

| Approach | Uses $f$ | Uses $g$ | Uses $\mathcal{L}_{train}$ |
|---|---|---|---|
| Random | ✗ | ✗ | ✗ |
| PCA | ✗ | ✓ | ✗ |
| Low-rank $W_{l+1}$ | ✓ | ✗ | ✗ |
| Jacobian | ✓ | ✓ | ✗ |
| Optimized Basis | ✓ | ✓ | ✓ |

Table 1: Design space of the different bases and approaches considered in the paper. $g$ is the network encoder (till layer $\ell$) and $f$ is the downstream classifier as illustrated in fig. 2. $\mathcal{L}_{train}$ is the training loss.

**Function Sensitivity**: The Jacobian matrix encodes the sensitivity of the classification function $f$ to varying input features $\mathbf{z}_i$. We derive a global feature set (independent of a particular $\mathbf{z}_i$) via the columns of the Jacobian. Intuitively, this gives us a set of directions which maximally explains the sensitivity of the function in the space $V$. Concretely, we consider the matrix $G$ of samples $\{\frac{df(\mathbf{z}_i)_j}{d\mathbf{z}_i}\}_{i=1}^n$, where each sample is the Jacobian vector of one output logit $j \in \mathcal{Y}$ (chosen by sampling from the output distribution of the classifier $f$). We do this since backpropagating through each logit (for each training datapoint) in turn is too expensive. Next, we decompose $G = U \cdot S \cdot V^T$ using singular value decomposition (SVD), and use the columns $\mathbf{v}_i \in \mathbb{R}^K$ of the right-singular matrix $V$ as the feature set. This gives us a set of directions in the input space which explain the dominant directions in which the function varies (but does not capture directions which necessarily have variance in the training dataset $\mathcal{D}_{train}$).

**Random Basis:** We start with $V \sim \mathcal{N}(0, \mathbf{I})$, where $\mathbf{I}$ is the identity matrix in $\mathbb{R}^{K \times K}$ and then use Gram-Schmidt orthonormalization to get a full rank random matrix $V$ for truncating features $V_m$.

**Low Rank Linear Layer:** Consider the linear embedding (or convolutional kernel) $W_{l+1}$ in the layer $l + 1$ proceeding $\mathbf{z} = g(\mathbf{x})$. We consider here low-rank approximations to this weight matrix to study if any potential OoD improvements come from a nullspace phenomenon or some other kind of capacity control. Following (Kolda & Bader, 2009), we consider the SVD of $W_{l+1} = A \cdot S \cdot B^T$, where S is a diagonal matrix of singular values $\text{diag}([s_1, \cdots, s_r])$ such that $s_1 \geq s_2 \geq \cdots \geq s_r$, and $r$ is the rank. A low-rank approximation to $W_{l+1}$ can then be easily found by truncating $s$ to top

$m \leq r$ singular values, that is, $S_m = \text{diag}([s_0, \cdots, s_m, 0, \cdots, 0])$ and the corresponding low rank $W_{l+1}^m = A \cdot S_m B^T$.

## 3 EXPERIMENTAL SETUP

**Training Protocol:** We evaluate on a number of domain generalization datasets provided as a part of the DomainBed (Gulrajani & Lopez-Paz, 2020) benchmark. Specifically, we consider RotatedMNIST (Ghifary et al., 2015), PACS (Li et al., 2017b), TerraIncognita (Beery et al., 2018), DomainNet (Peng et al., 2019), and Imagenet-R (Hendrycks et al., 2021). We train ERMs using the hyperparameter selection strategy pioneered by (Gulrajani & Lopez-Paz, 2020), where we train a large number of ERMs using random hyperparameter search, pick the best one on IID validation and use it to report OOD test accuracy – repeating this process for multiple independent trials. On RotatedMNIST we consider two differences to the standard DomainBed protocol: 1) we train on individual domains and evaluate on all others (to emphasize the fact that our method conceptually does not need environment annotations), and 2) we include a search over weight decay (which is fixed to 0 for RotatedMNIST in (Gulrajani & Lopez-Paz, 2020)) since weight decay might be related to smoothness of $f$ and nullspace occupancy. Following DomainBed, we use ResNet50 for all experiments on PACS, TerraIncognita, DomainNet, and ImageNet-R, and a small CNN for experiments on Rotated-MNIST. We note that both DomainBed and ImageNet-R are released under MIT license. To our knowledge, none of these datasets contain personally identifiable information or offensive content.

**OOD Accuracy Oracle:** Given trained networks $f \cdot g$ (fig. 2), our objective is to validate whether OOD accuracy improves when we project out the nullspace components for a test data point. Each of the datasets above has multiple train and test environment splits in DomainBed. For the best ERM network that DomainBed yields for a given set of training evironments, we sweep through all the layers and for each layer, we sweep through different values of $m$, perform the projection and note the peak performance improvement achieved. We then average this number across multiple choices of the training environments (each of which corresponds to a different network) to obtain the "oracle" accuracy (**o**) improvement. Following DomainBed methodology, we repeat the entire process for 3 independent trials, and report the mean oracle accuracy achieved. In another experiment, we sweep through all the layers, but choose the value of $m$ via. some heuristic that can be computed via. $\mathcal{D}_{train}$. In such cases we call the oracle improvement as "layer-oracle" accuracy improvement or **lo**, since it is an oracle that has access to the ideal layer. The OOD accuracy oracle provides an upper bound to the improvement achievable by our feature projection method, and is indicative of the amount of generalization failure explained by nullspace failure.

**Leave-one-out evaluation:** A limitation of the oracle evaluation method is that there is no way to pick the correct layer in which to perform projection. Towards obtaining a practically achievable improvement, we perform an additional experiment where we select the layer $l$ and number of components $m$ based on highest accuracy on a held-out training domain. Our methodology for PACS, TerraIncognita, and DomainNet datasets is as follows: given a test domain, we hold out one training domain for selecting projection hyperparameters and train a network using ERM on the remaining training domains. We fit bases on the remaining domains, select the layer $l$ and number of components $m$ to get the highest achievable improvement on the held-out domain, and evaluate performance on the test domain. We train a network on every combination of training and held-out domain, and compute the mean accuracy on the test domain over networks trained with different held-out domains. We repeat this entire process for 3 independent trials, and report mean accuracy and standard error. For Rotated-MNIST, we train networks on 0-degree data, select the layer and number of components on 45-degree data, and evaluate on remaining domains.

## 4 RESULTS

**Does nullspace occupancy relate to OoD failure?** We first illustrate (fig. 3) how the different choices of bases affect accuracy of the pretrained ERM models when performing projection (for a randomly chosen network and an oracle choice of layer for the RotatedMNIST, PACS, and TerraIncognita datasets, see Appendix B for a similar plot on DomainNet). First, notice that projecting with a random

Figure 3: **Nullspace occupancy relates to OoD failure:** Projecting test OoD activations onto a subspace of rank $m$ (x-axis) improves OoD performance (y-axis). For each dataset – Rotated-MNIST (left), PACS (middle), and TerraIncognita (right) – we pick one illustrative network out of 12 networks and the layer which best explains the nullspace occupancy related failure mode.

basis (green) does not improve performance over the base ERM. Instead, the performance converges montonically to that of the base ERM model when $m$ is full-rank.

In contrast to the behavior of the random basis, methods which utilize $f$, $g$ or $\mathcal{L}_{train}$ (table 1) exhibit a different structure with respect to $m$. While they all converge to the base ERM accuracy at full rank $m$ (right end of the plot), at intermediate values of $m$ we observe that projection improves the OoD performance of the network. More specifically, with increasing number of components, the performance first increases to an optimal choice of $m$ and then decreases as more and more spurious nullspace components are added.

The best OoD performance is achieved by the optimized basis (red line fig. 3), which explicitly finds the directions leading to the highest marginal improvement in the training loss $\mathcal{L}_{train}$ and discards directions which don't improve $\mathcal{L}_{train}$. This suggests that the optimized basis finds a more meaningful set of directions for a given $m$ compared to utilizing PCA (that only considers $g$ and not how $f$ uses the different directions) or the Jacobian-based feature spaces (which considers $f$'s sensitivity to different directions but not whether the sensitivity contributes to improvement in the training loss $\mathcal{L}_{train}$). Overall, this suggests that nullspace occupancy is a mechanism of OoD failure, and that the optimized basis is the best choice of basis $V$ with which to investigate this mechanism.

**Does projection improve OoD performance?** Next we study what improvements in performance are possible with perfect heuristics or "oracle" choices of layer $l$ and rank $m$ on the DomainBed benchmark. In addition to ERM, we compare with the CORAL (Sun & Saenko, 2016), SagNet (Nam et al., 2019), and Interdomain-Mixup (Yan et al., 2020) models (from DomainBed) which achieve state-of-the-art performances on the datasets explored in our work.

*Heuristic for choosing $m$:* As explained in (section 3) we report two oracles for each basis: 1) "oracle" assumes access to both the best layer as well as best number of components $m$, and 2) "oracle-layer" which selects $m$ using a heuristic and picks the best layer. Thus, "oracle-layer" is a more practical indicator of achievable OoD improvements while "oracle" provides an upper bound on achievable performance. For the "oracle-layer" approach we choose $m$ at the point when the training accuracy saturates to $99.9\%$ of the full value.

Notice that for the optimized basis, the oracle numbers (**o**) compare favourably to the SagNet and CORAL algorithms on PACS, TerraIncognita as well as DomainNet (suggesting possible improvements of $3.3 \pm 1.0$ over SagNet on TerraIncognita, for example). Interestingly, the layer oracle (**lo**) which uses a heuristic for choosing $m$ also gives consistent improvements over ERM (table 2). A similar table with results on in-distribution generalization performance is included in Appendix F.

One concern when reporting oracle numbers as an aggregate, maximizing over $m$ and choosing an oracle layer $L$, is if the reported improvements are significant or whether even a baseline model (at the same performance as the original ERM equipped with randomness in how it labels) could achieve the same improvements given $m \times L$ trials. We compare against such baseline models (for the multi-class classification case) in Appendix C. Our results show that the systematic improvements we report in table 2 are highly unlikely to occur due to random chance (see Appendix C for details).

**Does nullspace occupancy yield non-oracle improvements?** We next study if it is possible to choose the layer $l$ and rank $m$ using an unseen training domain, to obtain performance improvements on a novel, held-out OoD test domain. Concretely, we train a network using standard DomainBed protocol on say $d$ training domains, use an unseen training domain $d + 1$ to pick $l$ and $m$, and

Table 2: Aggregate domain generalization performance achieved by different bases on RotatedMNIST, PACS, TerraIncognita, and DomainNet. For each basis, we report the layer-oracle accuracy (**lo**) and the overall oracle performance (**o**). Error bars are in Appendix A along with per-domain breakdowns.

| | RotatedMNIST | | PACS | | TerraIncognita | | DomainNet | | Imagenet-R | |
|---|---|---|---|---|---|---|---|---|---|---|
| ERM | 62.8 | | 85.9 | | 48.7 | | 40.9 | | 36.1 | |
| **ERM + Different Low-Rank bases $V_m$:** | | | | | | | | | | |
| | RotatedMNIST | | PACS | | TerraIncognita | | DomainNet | | Imagenet-R | |
| | **lo** | **o** | **lo** | **o** | **lo** | **o** | **lo** | **o** | **lo** | **o** |
| Random | 62.8 | 63.2 | 85.6 | 86.4 | 48.7 | 49.3 | 40.7 | 40.9 | 36.1 | 36.1 |
| Jacobian | 62.9 | 63.7 | 86.6 | 87.1 | 49.3 | 50.6 | 41.0 | 41.6 | 37.2 | 38.2 |
| PCA | 63.0 | 64.3 | 86.9 | 87.4 | 50.1 | 51.6 | 41.4 | 42.2 | 37.1 | 37.4 |
| Low Rank $W_{l+1}$ | 62.8 | 63.4 | 85.9 | 86.7 | 48.7 | 50.0 | 41.1 | 41.6 | - | - |
| Optimized | **63.5** | **64.8** | **87.4** | **88.1** | **50.5** | **51.9** | **41.6** | **42.4** | **37.4** | **38.4** |
| **Representative state-of-the-art approaches:** | | | | | | | | | | |
| CORAL | N/A | | $86.2 \pm 0.3$ | | $47.6 \pm 1.0$ | | $41.5 \pm 0.1$ | | - | |
| SagNet | N/A | | $86.3 \pm 0.2$ | | $48.6 \pm 1.0$ | | $40.3 \pm 0.1$ | | - | |
| Mixup | N/A | | $84.6 \pm 0.6$ | | $47.9 \pm 0.8$ | | $39.2 \pm 0.1$ | | - | |

Table 3: Aggregate domain generalization performance achieved by different bases on RotatedMNIST, PACS, and TerraIncognita using leave-one-out hyperparameter selection. We report mean results and standard errors over three independent trials.

| | RotatedMNIST | PACS | TerraIncognita |
|---|---|---|---|
| ERM | $65.6 \pm 0.7$ | $78.2 \pm 0.8$ | $39.3 \pm 1.1$ |
| **ERM + Different Low-Rank bases $V_m$:** | | | |
| Random | $64.6 \pm 1.3$ | $76.4 \pm 1.6$ | $34.8 \pm 2.3$ |
| Jacobian | $65.4 \pm 0.7$ | $78.1 \pm 0.8$ | $38.1 \pm 2.4$ |
| PCA | $65.7 \pm 0.6$ | $78.1 \pm 1.3$ | $38.9 \pm 1.3$ |
| Optimized | $\mathbf{66.3 \pm 0.6}$ | $\mathbf{79.1 \pm 1.2}$ | $\mathbf{39.6 \pm 1.4}$ |

report performance after projection on a test domain $d + 2$. Encouragingly, the optimized basis yields consistent improvements over ERM across all three datasets we evaluated on, namely, Rotated-MNIST, PACS, and TerraIncognita (table 3). Further, the optimized basis is the only method that achieves consistent improvements over ERM in this setting. This demonstrates the value of learning the optimized basis that makes use of $f$, $g$ and $\mathcal{L}$ (table 1), over other baseline approaches for choosing the basis. Importantly, this also indicates that reducing nullspace occupancy is a viable method for improving OoD performance of already trained ERM networks in more practical, non-oracle settings.

**Is nullspace occupancy distinct from "capacity control"?** We compare against keeping the same activations $\hat{\mathbf{z}}$ in layer $l$ at test time, but instead reduce the rank of the matrix $W_{l+1}$ in the next layer. This implements a simple, inference-time way to control the capacity of the function $f$ locally around layer $l$. Across all datasets, for both oracle as well as layer oracles table 2, we find that this yields inferior results to both the optimized basis as well as the PCA basis. This result demonstrates that it is important to take both the variance in the training data, as well as the overall behavior of $f$ into account when reducing the rank, as opposed to reducing the rank only based on local characteristics of the network in the next layer.

**Is projection complementary to L2 regularization?** L2 regularization (Krogh & Hertz, 1991) is a commonly used heuristic for training deep models and is thought of commonly as a regularizer. Given parameters $\theta$ and a loss function $\mathcal{L}_{train}$, L2 regularization optimizes $\mathcal{L}_{train} + \lambda ||\theta||_2^2$, where $|| \cdot ||$ denotes the $L^2$ norm. It is intuitive that L2 regularization should lead to "simpler" networks (in the linear case this corresponds to lower-rank solutions) which may not suffer from the nullspace failure mode. While the results in table 2 already include models trained with weight decay (based on standard DomainBed hyperparameter sweeps), in this section we further scrutinize the connection between nullspace occupancy failure and L2 regularization, since it is possible that in the presence of such regularization one might extrapolate more smoothly in a low-rank nullspace (see appendix E).

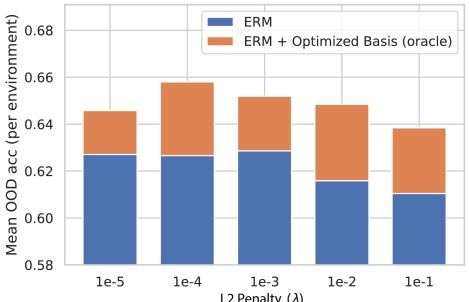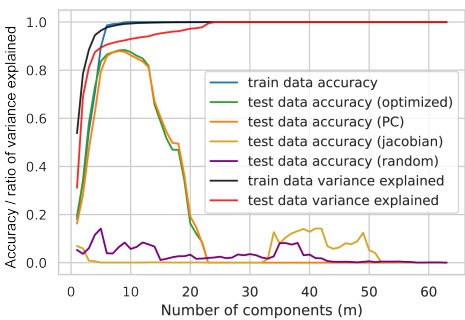

Figure 4: **(left) Oracle improvements (o) vs L2 penalty:** For Rotated-MNIST, for each bin of L2 penalty values and test domain, we train 15 networks and select the one that achieves highest validation accuracy. We plot the OoD accuracy of selected networks averaged across test domains along with the average oracle improvement on the selected networks. We observe our method give comparable performance for all L2 penalty values. **(right) Nullspace occupancy for poisoned test data:** Classification accuracy after projecting network activations to the top-m components in the PC basis and random basis. The black and red curves are the amount of total variance explained by the first-m components. We observe that for PC basis, test set accuracy increases till training variance explained is saturated and sharply declines when including nullspace components.

Instead of the standard DomainBed procedure, where for each set of train and test environments (termed a condition) one trains a large number of ERM models via random hyperparameter selection (Gulrajani & Lopez-Paz, 2020) and picks the best model, we instead bin the networks into sets based on the strength $\lambda$ of the L2 penalty used to train them (fig. 7) and then pick the best ERM model (using validation data from the training domain). This allows us to assess the complementary benefits of our approach (which completely projects down to a chosen subspace) over L2 regularization which should in principle ensure more smooth functions.

We plot performance of the best model in each L2 penalty bin and compare it to the oracle improvements achievable over that specific model using projection into the optimized basis (fig. 7) on RotatedMNIST (see Appendix B for more datasets). We find that projection gives consistent improvements over L2 regularization across the different buckets (x-axis fig. 7, right). This suggests that L2 regularization and projection are quite complementary to each other, and that nullspace failure occurs in deep networks even in the presence of L2 regularization.

**Does nullspace occupancy emerge in data poisoning?** So far we studied a setup where we intervened by projecting the representation down to a subspace $V_m$, and we observed that this can improve the performance of the network $f$ operating on the projected features $\mathbf{v}_m^\mathbf{x}$. What if we instead did the reverse? That is, does the nullspace occupancy phenomenon *emerge* in cases where we explicitly force neural networks to not generalize?

To test this, we utilize the experimental setting of (Huang et al., 2020) who show how to take training and test datasets $\mathcal{D}_{train} = \{\mathbf{x}_i, y_i\}_{i=1}^{|\mathcal{D}_{train}|}$ and $\mathcal{D}_{test} = \{\mathbf{x}_i, y_i\}_{i=1}^{|\mathcal{D}_{test}|}$ and find a network $u_\theta$ that generalizes poorly to $\mathcal{D}_{test}$. Let $u_\theta : \mathcal{X} \to \mathcal{P}(\mathcal{Y})$ be the neural network mapping inputs $\mathcal{X}$ to the a distribution over the labels $\mathcal{P}(\mathcal{Y})$. Then we train to optimize:

$$\min_\theta \frac{1}{|\mathcal{D}_{train}|} \sum_{i=1}^{|\mathcal{D}_{train}|} -\log u_\theta(y_i|\mathbf{x}_i) + \frac{1}{|\mathcal{D}_{test}|} \sum_{i=1}^{|\mathcal{D}_{test}|} -\log(1 - u_\theta(y_i|\mathbf{x}_i)) \qquad (4)$$

We perform our experiment on the MNIST dataset by splitting the dataset into a train and test split. We essentially train to perform well on the training set and badly on the test set, yielding a small convolutional network $u_\theta$ which by construction performs poorly on the given test set $\mathcal{D}_{test}$. Having "poisoned" the test set, we check if projecting the test datapoints $\mathbf{x} \in \mathcal{D}_{test}$ to a subspace $V_m$ can "de-poison" the network. fig. 4 shows that this indeed happens in layer 5 for this network, and that one can recover a large fraction of the lost performance due to poisoning by projecting into the first $m$ components of the PCA basis that explain most of the variance.

This indicates that the phenomenon of nullspace occupancy contributing to generalization failure has applicability in other settings which are not necessarily connected to out-of-distribution general-

ization and hints at some interesting underlying mechanism that connects nullspace occupancy to generalization more broadly. See Appendix G for additional experimental details.

Finally, we performed experiments on adversarial robustness (see appendix K) where we found similar trends as data poisoning – namely that both PCA as well as the optimized basis lead to gains in robustness, but the gap between PCA and optimized basis is not as large as observed in more traditional OoD generalization tasks (such as on DomainBed).

## 5 RELATED WORK

**Out-of-Distribution (OoD) Generalization.** There has been considerable interest in the out-of-distribution generalization problem (Gulrajani & Lopez-Paz, 2020; Arjovsky et al., 2019; Sagawa et al., 2019; Li et al., 2017a) where the goal is to generalize the classifier $f \cdot g$ to novel input domains $\mathcal{D}$. A number of popular approaches have been pursued including causal invariance (Arjovsky et al., 2019), gradient starvation (Shah et al., 2020; Pezeshki et al., 2020), risk extrapolation (Krueger et al., 2020), meta-learning (Li & Li, 2017), distributionally robust optimization (Sagawa et al., 2019; Sinha et al., 2017) *etc.*. Among these, causal invariance and gradient starvation (Pezeshki et al., 2020) (or simplicity bias (Shah et al., 2020)) are quite relevant to us, since they prescribe certain failure modes which might be causing poor out-of-distribution accuracy. In contrast to work on causal invariance, our work does not require an underlying "true" causal model or annotation of different environments (except for the purposes of leave-one-out domain evaluation). While gradient starvation is concerned with features $\mathbf{v}_1$ and $\mathbf{v}_2$ where both are potentially useful, our work is focused on removal of additional/extra features, rather than using all the relevant features. In this sense, gradient starvation and our work are quite complementary. Another relevant approach is that of last-layer feature reweighting. Rosenfeld et al. (2022) and Kirichenko et al. (2022) show that ERM already learns some generalizable features by showing that retraining the last layer using access to the target distribution or a non-spuriously correlated reference distribution improves OOD performance. In contrast, we show that it is possible to drop extra nullspace features without access to a target or reference distribution to improve network performance.

**Nullspaces, PCA, and Representation Learning.** Low-rank feature spaces (primarily using PCA) have been utilized for detecting adversarial examples (Li & Li, 2017; Carlini & Wagner, 2017) and for improving adversarial robustness (Sanyal et al., 2018). In contrast to these works, we are focused on "semantic" OoD generalization as opposed to adversarial examples. Other recent work has noted that neural networks learn feature spaces which are essentially low rank (Huh et al., 2021). Our work makes use of that observation to study OoD generalization and how choices of bases affect generalization. In contrast to these works, which largely utilize PCA, our work considers the downstream network $f$ as well to identify the directions that are most predictive of the label on the training set. In this sense, our work is like a deep version of partial least squares (PLS) (Geladi & Kowalski, 1986) which aims to find the discriminative directions in feature space for linear models.

**Low-rank models and Pruning.** There is a lot of work in model compression that focusses on reducing the rank of the layer weights $W_l$ after training (Cichocki et al., 2016; Udell et al., 2016). A naive data-independent strategy includes minimizing the distance $||W - UV^T||$ between the original network layer weight $W$ and the compressed weights $UV^T$, using a closed form solution such as SVD Denton et al. (2014); Novikov et al. (2015). We utilize these techniques in our low-rank $W_{l+1}$ baseline (section 2) to compare against our feature projection techniques (table 2). In contrast to these works which focus on compression, our goal is instead to improve downstream OoD performance (our low-rank basis does not yeild any model compression). (Zhang et al., 2021) relates model pruning to out-of-distribution generalization, showing that even in models which rely on spurious correlations, there exist subnetworks that are robust to spurious correlations. Similar to this work, we reduce the capacity of the network (or a particular layer in our case) to study OoD generalization. However, unlike network pruning, our technique focuses on a particular layer and exposes nullspace occupancy as a mechanism by which OoD failure can happen.

## 6 DISCUSSION

**Nullspace removal is not sufficient for improving generalization.** We observe that while there usually exists a layer $l \in \{1, \cdots, L\}$ where performing the projection improves the oracle out of

distribution accuracy, it is not necessary that the projection helps in every layer. To understand this, it is helpful to think of a breakdown of error $\epsilon$ into two different sources, namely $\epsilon_{null}$ which is the nullspace error and $\epsilon_{ds}$ which is the distribution shift error. One can have a distribution shift even if the test data spans the top $V_m$ components which explain the training data very well since even if the components are same, the distribution of the test datapoints could end up being quite different. Thus, it is not necessary that removing the nullspace error $\epsilon_{null}$ would lead to a low overall error $\epsilon$. However, in practice it seems to play a role in OoD generalization.

**Nullspace occupancy is not necessary for poor generalization.** It is also straightforward to notice that, if a trained network learns appropriately smooth functions $f(\mathbf{v}_1, \mathbf{v}_2)$, ignoring an extra feature $\mathbf{v}_3 = 0$ for example, then it would not be troubled by test data occupying the nullspace. In practice, it appears that networks learnt even in the presence of weight decay do not show such a behavior which suggests that nullspace occupancy might be a mechanism for explaining generalization failure in those cases.

**Connection to other OoD generalization failure modes.** Given input features $\{\mathbf{v}_1, \mathbf{v}_2\} \in \mathcal{V}$ present in a dataset $\mathcal{D}_{train}$, and a trained neural network function $f(\mathbf{v}_1, \mathbf{v}_2)$ from $\mathcal{V} \to \mathcal{Y}$, where $\mathcal{Y}$ is the label space (fig. 1), recent work has studied two popular modes of out-of-distribution generalization failure with deep learning:

- **Causality:** Given a label $y$, a causal feature $\mathbf{v}_1$ and a "spurious" correlated feature $\mathbf{v}_2$, the idea is to learn a function $f(\mathbf{v}_1)$ that ignores $\mathbf{v}_2$ (Arjovsky et al., 2019). Instead, if we learn a function $f(\mathbf{v}_2)$, we fail due to spurious correlation.
- **Gradient Starvation (Pezeshki et al., 2020; Shah et al., 2020):** Given two predictive features $\mathbf{v}_1$ and $\mathbf{v}_2$, where $\mathbf{v}_1$ elicits a stronger response at initialization than $\mathbf{v}_2$, the dynamics of learning yields $f(\mathbf{v}_1)$ that ignores the weaker, but predictive feature $\mathbf{v}_2$. This can lead to generalization failure in a novel environment.

Our approach is distinct from causality in the sense that we are simply measuring geometric properties of the learnt features as opposed to assuming any underlying causal model that induces OoD failure. More closely related to our work is gradient starvation. In the language of gradient starvation, let us assume a stronger feature $\mathbf{v}_1$, a less dominant feature $\mathbf{v}_2$ and an invariant feature $\mathbf{v}_3 = 0$. Gradient starvation considers the case of $\mathbf{v}_1$ and $\mathbf{v}_2$, and suggests that the strong feature $\mathbf{v}_1$ might inhibit the learning of the weak feature $\mathbf{v}_2$, at least in the neural tangent kernel regime (Jacot et al., 2018). A priori, one might expect that $\mathbf{v}_1$ would suppress the learning of $\mathbf{v}_3$ as well, but our work shows empirically that this is not the case. Our initial attempts to extend the theory of (Pezeshki et al., 2020) revealed that their framework cannot easily accommodate a notion of a "useless" feature, and the linearity assumptions in the NTK regime might not explain the non-smooth behavior of deep neural networks (see Appendix D). However, it is interesting to note that while the gradient starvation work attempts to mitigate this and learn more features, our work in contrast attempts to discard uninformative and redundant features.

## 7 CONCLUSION

In this work we introduced the concept of nullspace occupancy – namely when test data occupies the nullspace of a low-rank feature space $V_m$ that captures the training set variability – and connected it to OoD generalization. We showed that with a careful choice of the basis $V_m$, systematic improvements in OoD generalization can often be obtained by projecting out the remaining nullspace components for test data points. We found this "optimized basis" by performing an optimization over orthogonal matrices to identify a set of components that minimizes the area under the training loss curve for different choices of rank $m$. On domain generalization experiments on RotatedMNIST, PACS and TerraIncognita we found that this choice of basis has the potential to improve out-of-distribution accuracy of ERMs. Finally, we also found that nullspace occupancy emerges in a setting where one poisons the network to perform poorly on a predetermined test set. Together, these results hint at a broader interplay between nullspace occupancy and generalization in deep learning. There is much to explore to understand the ubiquity of this novel mode of generalization failure and further refine our approach to mitigate it, and we hope the community works to address these issues in the future.

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

We include the following appendices:

- Appendix A: We report per-domain domain generalization results for projection in all bases on Rotated-MNIST, PACS, TerraIncognita, and DomainNet.

- Appendix B: We provide figures from the main text on additional datasets.

- Appendix C: We demonstrate that our reported oracle results are not an artifact of considering the maximum of many models. To establish this, we compare achieved improvements against the maximum performance achieved by a large set of models with the same accuracy as the base ERM, equipped with randomness.

- Appendix D: We discuss the connection between nullspace components and the theoretical framework of Gradient Starvation (Pezeshki et al., 2020).

- Appendix E: We include a formal discussion of weight decay and how it relates to the nullspace phenomenon

- Appendix F: We report results for projection on in-distribution generalization for Rotated-MNIST, PACS, TerraIncognita, and DomainNet.

- Appendix G: We report additional details for the data poisoning experiment.

- Appendix H: We interpolate between test datapoints $\hat{z}$ (with components in the nullspace) to the projected test datapoints $\hat{z}^m$ and study how accuracy varies as we move between them

- Appendix I: We present the small convolutional architecture used for all experiments on Rotated-MNIST.

- Appendix J: We include plots of out-of-distribution accuracies after projection into all the different bases, for all networks we consider across every dataset.

- Appendix K: We study the extent to which the optimized basis can help defend against adversarial attacks.

- Appendix L: We include pseudocode for computing the oracle and layer-oracle accuracies reported in the paper.

- Appendix M: We include pseudocode for obtaining our optimized basis by optimizing equation 3.

## A    PER-DOMAIN DOMAIN GENERALIZATION RESULTS

In this section, we show per-domain results of Table 2 from the main text. For each test domain, we report the mean accuracy and SEM over three nets selected using DomainBed search protocols, after projection into different low rank feature spaces.

Table 4: Per-domain oracle (**o**) and layer-oracle (**lo**) out-of-distribution classification accuracies for different choices of bases on Rotated-MNIST.

| Algorithm | 0 | 15 | 30 | 45 | 60 | 75 | Avg |
|---|---|---|---|---|---|---|---|
| ERM | $100.0 \pm 0.0$ | $96.4 \pm 0.0$ | $79.1 \pm 1.1$ | $49.9 \pm 1.8$ | $31.6 \pm 2.0$ | $19.9 \pm 1.0$ | 62.8 |
| Random (**o**) | $100.0 \pm 0.0$ | $96.5 \pm 0.0$ | $79.7 \pm 1.6$ | $50.4 \pm 1.6$ | $32.1 \pm 1.5$ | $20.7 \pm 2.1$ | 63.2 |
| Jacobian (**o**) | $100.0 \pm 0.0$ | $96.7 \pm 0.1$ | $80.2 \pm 1.4$ | $50.8 \pm 2.1$ | $32.3 \pm 1.9$ | $22.0 \pm 0.7$ | 63.7 |
| PCA (**o**) | $100.0 \pm 0.1$ | $96.9 \pm 0.1$ | $80.9 \pm 1.2$ | $51.6 \pm 1.4$ | $34.6 \pm 1.8$ | $21.6 \pm 0.5$ | 64.3 |
| Low Rank $W_{l+1}$ (**o**) | $100.0 \pm 0.0$ | $96.7 \pm 0.0$ | $79.6 \pm 1.1$ | $50.3 \pm 1.8$ | $32.6 \pm 1.5$ | $21.4 \pm 0.7$ | 63.4 |
| Optimized (**o**) | $100.0 \pm 0.0$ | $\mathbf{97.0 \pm 0.1}$ | $\mathbf{81.0 \pm 1.2}$ | $\mathbf{52.4 \pm 1.2}$ | $\mathbf{36.1 \pm 1.6}$ | $\mathbf{22.0 \pm 1.0}$ | **64.8** |
| Random (**lo**) | $100.0 \pm 0.0$ | $96.4 \pm 0.0$ | $79.1 \pm 1.1$ | $49.9 \pm 1.8$ | $31.6 \pm 2.0$ | $20.0 \pm 1.0$ | 62.8 |
| Jacobian (**lo**) | $99.9 \pm 0.0$ | $96.4 \pm 0.0$ | $79.3 \pm 1.4$ | $49.0 \pm 2.2$ | $31.8 \pm 2.1$ | $20.8 \pm 1.2$ | 62.9 |
| PCA (**lo**) | $99.9 \pm 0.0$ | $96.4 \pm 0.1$ | $79.2 \pm 1.0$ | $50.7 \pm 1.2$ | $31.7 \pm 1.7$ | $20.4 \pm 0.8$ | 63.0 |
| Low Rank $W_{l+1}$ (**lo**) | $100.0 \pm 0.0$ | $96.5 \pm 0.0$ | $79.0 \pm 1.1$ | $49.9 \pm 2.0$ | $31.3 \pm 1.6$ | $19.8 \pm 1.0$ | 63.4 |
| Optimized (**lo**) | $99.9 \pm 0.0$ | $96.4 \pm 0.1$ | $79.7 \pm 1.1$ | $49.4 \pm 1.2$ | $35.2 \pm 1.3$ | $20.3 \pm 1.0$ | 63.5 |

Table 5: Per-domain oracle (**o**) and layer-oracle (**lo**) out-of-distribution classification accuracies for different choices of bases on PACS.

| Algorithm | A | C | P | S | Avg |
|---|---|---|---|---|---|
| ERM | $87.9 \pm 1.2$ | $81.4 \pm 1.7$ | $96.1 \pm 0.5$ | $78.0 \pm 1.7$ | 85.9 |
| Random (**o**) | $87.9 \pm 1.2$ | $81.8 \pm 1.0$ | $96.3 \pm 0.6$ | $79.7 \pm 1.4$ | 86.4 |
| Jacobian (**o**) | $88.6 \pm 1.2$ | $83.6 \pm 1.6$ | $96.4 \pm 0.6$ | $79.7 \pm 1.6$ | 87.1 |
| PCA (**o**) | $88.9 \pm 0.9$ | $83.7 \pm 1.6$ | $96.2 \pm 0.5$ | $80.7 \pm 1.0$ | 87.4 |
| Low Rank $W_{l+1}$ (**o**) | $88.5 \pm 0.7$ | $83.2 \pm 1.6$ | $96.1 \pm 0.5$ | $79.0 \pm 2.2$ | 86.7 |
| Optimized (**o**) | $\mathbf{89.1 \pm 1.0}$ | $\mathbf{84.2 \pm 1.8}$ | $\mathbf{96.7 \pm 0.3}$ | $\mathbf{82.4 \pm 0.3}$ | **88.1** |
| Random (**lo**) | $86.9 \pm 1.1$ | $81.0 \pm 0.8$ | $96.3 \pm 0.6$ | $78.1 \pm 1.4$ | 85.6 |
| Jacobian (**lo**) | $87.9 \pm 1.1$ | $82.9 \pm 1.4$ | $96.2 \pm 0.6$ | $79.4 \pm 0.6$ | 86.6 |
| PCA (**lo**) | $88.5 \pm 1.0$ | $82.9 \pm 1.7$ | $96.2 \pm 0.5$ | $80.1 \pm 1.3$ | 86.9 |
| Low Rank $W_{l+1}$ (**lo**) | $87.9 \pm 1.2$ | $81.4 \pm 1.7$ | $96.1 \pm 0.5$ | $78.3 \pm 1.7$ | 85.9 |
| Optimized (**lo**) | $88.0 \pm 0.9$ | $82.9 \pm 1.9$ | $96.5 \pm 0.4$ | $82.0 \pm 0.5$ | 87.4 |

Table 6: Per-domain oracle (**o**) and layer-oracle (**lo**) out-of-distribution classification accuracies for different choices of bases on TerraIncognita.

| Algorithm | L100 | L38 | L43 | L46 | Avg |
|---|---|---|---|---|---|
| ERM | $53.6 \pm 3.5$ | $43.9 \pm 1.2$ | $56.8 \pm 1.5$ | $40.4 \pm 1.2$ | 48.7 |
| Random (**o**) | $54.5 \pm 3.7$ | $44.7 \pm 1.8$ | $57.4 \pm 1.9$ | $40.7 \pm 0.4$ | 49.3 |
| Jacobian (**o**) | $56.9 \pm 2.7$ | $46.0 \pm 1.0$ | $57.8 \pm 1.4$ | $41.7 \pm 1.2$ | 50.6 |
| PCA (**o**) | $58.1 \pm 2.1$ | $47.6 \pm 0.8$ | $58.7 \pm 1.7$ | $\mathbf{41.9 \pm 1.7}$ | 51.6 |
| Low Rank $W_{l+1}$ (**o**) | $55.8 \pm 2.2$ | $46.3 \pm 1.0$ | $57.0 \pm 1.6$ | $41.0 \pm 1.8$ | 50.0 |
| Optimized (**o**) | $\mathbf{58.6 \pm 2.2}$ | $\mathbf{47.9 \pm 2.4}$ | $\mathbf{59.3 \pm 1.3}$ | $41.9 \pm 1.1$ | **51.9** |
| Random (**lo**) | $53.5 \pm 0.3$ | $44.3 \pm 1.3$ | $57.2 \pm 2.1$ | $39.9 \pm 0.8$ | 48.7 |
| Jacobian (**lo**) | $55.1 \pm 3.1$ | $44.2 \pm 1.4$ | $57.2 \pm 1.5$ | $41.0 \pm 1.1$ | 49.3 |
| PCA (**lo**) | $57.0 \pm 2.0$ | $45.4 \pm 0.9$ | $57.0 \pm 1.6$ | $41.1 \pm 1.3$ | 50.1 |
| Low Rank $W_{l+1}$ (**lo**) | $53.6 \pm 3.6$ | $44.0 \pm 1.2$ | $56.8 \pm 1.5$ | $40.4 \pm 1.2$ | 48.7 |
| Optimized (**lo**) | $56.1 \pm 3.4$ | $46.9 \pm 2.0$ | $57.6 \pm 1.3$ | $41.1 \pm 1.1$ | 50.5 |

## B ADDITIONAL PLOTS

In this section, we present figures from the main paper evaluated on additional datasets.

In Figure 3 of the main paper, we show that nullspace accuracy relates to OOD failure by plotting the improvement in OOD classification accuracy against the dimensionality of the subspace used for many different bases for Rotated-MNIST, PACS, and TerraIncognita. Fig 5 shows the corresponding plot for one randomly selected network from DomainNet.

In Figure 4 (right) of the main paper, we verify that the effect of projection is complementary to that of weight decay by plotting the improvements obtained by projection along with the base network accuracy for networks trained with different values of weight decay on Rotated-MNIST. fig. 6 presents the corresponding result on PACS, where we again see that projection gives consistent improvements across different buckets of weight decay, suggesting that weight decay and projection are complementary to each other.

## C COMPARISON WITH MAXIMUM PERFORMANCE OF A LARGE NUMBER OF BASELINE MODELS EQUIPPED WITH RANDOMNESS

In the main text, we report oracle numbers that are obtained by maximizing over many choices $m$ of subspace dimensionality, and oracle layer $L$. These numbers are the maximum over many different trials, raising the concern that they may not be significant because even a baseline model with the

Table 7: Per-domain oracle (**o**) and layer-oracle (**lo**) out-of-distribution classification accuracies for different choices of bases on DomainNet.

| Algorithm | clip | info | paint | quick | real | sketch | Avg |
|---|---|---|---|---|---|---|---|
| ERM | $59.6 \pm 0.8$ | $19.9 \pm 0.3$ | $45.5 \pm 0.9$ | $12.2 \pm 0.3$ | $59.2 \pm 0.6$ | $49.2 \pm 0.7$ | 40.9 |
| Random (**o**) | $59.6 \pm 0.8$ | $19.9 \pm 0.3$ | $45.5 \pm 0.9$ | $12.2 \pm 0.3$ | $59.2 \pm 0.6$ | $49.2 \pm 0.7$ | 40.9 |
| Jacobian (**o**) | $60.3 \pm 0.9$ | $20.5 \pm 0.4$ | $46.3 \pm 0.9$ | $13.0 \pm 0.1$ | $59.9 \pm 0.6$ | $49.8 \pm 0.5$ | 41.6 |
| PCA (**o**) | $61.0 \pm 0.9$ | $\mathbf{21.0 \pm 0.4}$ | $46.4 \pm 0.9$ | $\mathbf{13.8 \pm 0.1}$ | $60.3 \pm 0.6$ | $50.8 \pm 0.5$ | 42.2 |
| Low Rank $W_{l+1}$ (**o**) | $59.9 \pm 0.8$ | $20.5 \pm 0.4$ | $45.8 \pm 0.9$ | $13.0 \pm 0.2$ | $59.8 \pm 0.5$ | $50.4 \pm 0.7$ | 41.6 |
| Optimized (**o**) | $\mathbf{61.1 \pm 0.9}$ | $21.0 \pm 0.4$ | $\mathbf{47.0 \pm 0.9}$ | $13.5 \pm 0.1$ | $\mathbf{61.1 \pm 0.7}$ | $\mathbf{51.0 \pm 0.8}$ | **42.4** |
| Random (**lo**) | $59.4 \pm 0.7$ | $19.7 \pm 0.2$ | $45.5 \pm 0.5$ | $12.0 \pm 0.2$ | $58.3 \pm 0.2$ | $49.2 \pm 0.7$ | 40.7 |
| Jacobian (**lo**) | $59.7 \pm 0.9$ | $20.0 \pm 0.4$ | $45.5 \pm 0.9$ | $12.5 \pm 0.2$ | $59.4 \pm 0.6$ | $49.3 \pm 0.5$ | 41.0 |
| PCA (**lo**) | $60.1 \pm 1.0$ | $20.2 \pm 0.3$ | $45.7 \pm 1.0$ | $12.7 \pm 0.3$ | $59.7 \pm 0.6$ | $49.9 \pm 0.5$ | 41.4 |
| Low Rank $W_{l+1}$ (**lo**) | $59.4 \pm 0.8$ | $20.1 \pm 0.3$ | $45.3 \pm 0.9$ | $12.5 \pm 0.2$ | $59.4 \pm 0.5$ | $49.6 \pm 0.5$ | 41.0 |
| Optimized (**lo**) | $60.7 \pm 0.9$ | $\mathbf{21.0 \pm 0.4}$ | $45.8 \pm 1.1$ | $12.9 \pm 0.3$ | $59.6 \pm 0.8$ | $49.7 \pm 0.8$ | 41.6 |

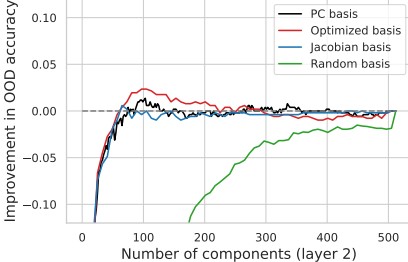

Figure 5: **Nullspace occupancy relates to OOD failure:** Projecting test OOD activations onto a subspace of rank $m$ (x-axis) improves OOD performance (y-axis). We randomly pick one of 18 networks trained on DomainNet and the layer which best explains the nullspace occupancy related failure mode.

same accuracy as ERM equipped with randomness in labelling could achieve the same improvements given $m \times L$ trials. In this section, we construct one reasonable choice of such a model, and compare the improvements obtained via our oracle method with an upper bound of the improvement obtainable via the random baseline model.

Let the base (ERM) model be $M$ with classification accuracy $x$ on the OOD test set. Let the number of samples in the test set be $n$, the number of layers $L$, and the number of considered choices of feature space dimensionality be $m$. In this setting, the effective number of experiments is $d = L \times m$. We therefore consider a set of non-deterministic models $W_1, \ldots, W_d$, where each model $W_i$ predicts the label correctly for each test sample with probability $x$. Let the out-of-distribution accuracies of models $W_1, \ldots, W_d$ be random variables $x_1, \ldots, x_d$. It follows that $\mathbb{E}[x_i] = x$, and $\mathbb{V}[x_i] = \frac{x(1-x)}{n}$.

We wish to compare the improvements obtained by our method with $\mathbb{E}\left[max_{i\in[d]}x_i\right]$. To obtain an upper bound for this expectation, we assume that variables $x_i \sim \mathcal{N}(x, \frac{x(1-x)}{n})$. We then use the following result:

**Theorem 1** *Orabona & Pál (2015) Let $X_1, \ldots, X_d$ be independent Gaussian Random Variables $N(0, \sigma^2)$. For any $d \geq 2$,*

$$\sigma\sqrt{2\log d} \geq \mathbb{E}\left[max_{i\in[d]}X_i\right] \geq \sigma\left(1 - exp\left(\frac{-\sqrt{\ln d}}{6.35}\right)\right)\left(\sqrt{2\ln d - 2\ln\ln d} + \sqrt{\frac{2}{\pi}}\right) - \sqrt{\frac{2}{\pi}}\sigma$$

This gives us the upper bound:

$$\mathbb{E}\left[max_{i\in[d]}x_i\right] \leq x + \sigma\sqrt{2\log d} = x + \sqrt{\frac{x(1-x)}{n}}\sqrt{2\log(mL)}$$

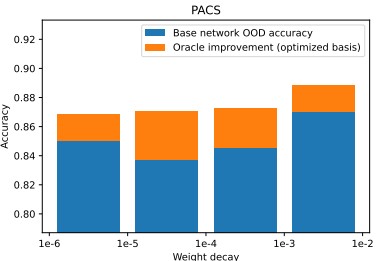

Figure 6: **Oracle improvements (o) vs weight decay:** For PACS, for each bin of weight decay values and test domain, we train 15 networks and select the one that achieves highest validation accuracy. We plot the OOD accuracy of selected networks averaged across test domains along with the average oracle improvement on the selected networks. We observe our method give comparable performance for all weight decay values.

We compare this upper bound with the oracle performance obtained via projection on the ImageNet-R dataset, which offers a tight upper bound due availability of a large number of test samples. In our experiments on ImageNet-R, we consider projections in $L = 5$ layers, and $m = 32$ choices of dimensionality of feature subspace. The base accuracy of ERM is $x = 36.1\%$, yielding an upper bound of $\mathbb{E}\left[max_{i \in [d]} x_i\right] - x \leq 0.88\%$. In contrast, the oracle accuracies of our projection method yield performance improvements of $1.3\%$ when using the PC basis, and $2.3\%$ when using the optimized basis. These results suggest that our method provides a systematic improvement to generalization performance, and is not an artifact of taking a maximum over a large number of models.

Note that the baseline model we consider, constructed by using the maximum performance with a set of models that achieve the same accuracy as the base model, is a very strong baseline. For reference, if we consider a naive model equipped with randomness that starts with the base ERM and randomly flips the prediction with probability $p$, it would achieve accuracy $p \cdot x + (1 - p)\frac{1-p}{\#classes}$. This could be significantly lower than $x$ and the maximum of a large number of such models could still yield performance less than $x$.

## D CONNECTION TO GRADIENT STARVATION (PEZESHKI ET AL., 2020)

Consider our running example of a training set with 3 features $\{\mathbf{v}_1, \mathbf{v}_2, \mathbf{v}_3 = 1\}$, where feature $\mathbf{v}_3$ is a nullspace feature, and has a constant value of 1. Gradient Starvation (GS) considers the setting where $\mathbf{v}_1$ is a "stronger" feature than $\mathbf{v}_2$, and suggests that the strong feature $\mathbf{v}_1$ inhibits the learning of the weak feature $\mathbf{v}_2$. In the context of our work, one might expect that $\mathbf{v}_1$ would also inhibit the learning of $\mathbf{v}_3$, thus potentially mitigating generalization failure due to nullspace occupancy. However, our empirical results indicate that this is not the case.

To reconcile this discrepancy, we attempt to fit nullspace features within the theoretical framework of Gradient Starvation. GS considers a setting with $n$ data points $\mathbf{X} = [\mathbf{x}_1, \dots, \mathbf{x}_n] \in \mathbb{R}^{n \times m}$, labels $\mathbf{y} = \{-1, 1\}^n$ and $\mathbf{Y} = \text{diag}(\mathbf{y}) \in \mathbb{R}^{n \times n}$, and function output $\hat{\mathbf{y}}(\mathbf{X}) = f(\mathbf{X})$ for neural network $f$. GS looks at the NTK parameterization, in which the network output is approximated as a linear model of the function of the parameters, near initial parameters $\boldsymbol{\theta}_0$:

$$\hat{\mathbf{y}}(\mathbf{X}, \boldsymbol{\theta}) = \hat{\mathbf{y}}(\mathbf{X}, \boldsymbol{\theta}_0) + \boldsymbol{\Phi}_0 \boldsymbol{\theta} \tag{5}$$

where $\boldsymbol{\Phi}_0 = \boldsymbol{\Phi}(\mathbf{X}, \boldsymbol{\theta}_0)$ is the neural tangent random feature (NTRF) matrix at initialization, i.e.: $\boldsymbol{\Phi}(\mathbf{X}, \boldsymbol{\theta}) = \frac{d\hat{\mathbf{y}}(\mathbf{X}, \boldsymbol{\theta})}{d\boldsymbol{\theta}} \in \mathbb{R}^{n \times m}$. GS then seeks to characterize the response of the network to a training example, where the response is defined as the deviation from its initial value:

$$\mathbf{r} = \mathbf{Y}(\hat{\mathbf{y}} - \hat{\mathbf{y}}_0) = \mathbf{Y}\boldsymbol{\Phi}_0 \boldsymbol{\theta}. \tag{6}$$

The theory studies features defined in terms of the singular value decomposition of the NTRF:

$$\mathbf{Y}\boldsymbol{\Phi}_0 = \mathbf{U}\mathbf{S}\mathbf{V}^T, \tag{7}$$

where $(\mathbf{V}^T)_{j\cdot}$ is the jth feature, $(\mathbf{S})_{jj}$ is the singular value of that feature and $(\mathbf{U})_{\cdot j}$ are the weights of the feature in each example adjusted according to the labels $y$. The network's response to each feature j can then be expressed as

$$\mathbf{z} = \mathbf{U}^T \mathbf{r} = \mathbf{S}\mathbf{V}^T \boldsymbol{\theta}. \tag{8}$$

To evaluate the prediction of the Gradient Starvation framework to nullspace features, our goal is to look at the response to nullspace features, when they appear in a new out-of-distribution example $x_o$. Note first that features considered under the framework are obtained via singular value decomposition of the NTRF, and the diagonal matrix of singular values $\mathbf{S}$ has rank $n$, i.e. all the singular values are positive. Hence, the Gradient starvation framework does not include any features with singular value zero (nullspace features), and can only explain suppression of features in the training subspace.

To attempt to extend the framework to include nullspace features, we extend the feature set obtained via SVD of the NTRF, by extending the set of examples $[\mathbf{x}_1, \ldots, \mathbf{x}_n, \mathbf{x}_o] \in \mathbb{R}^{(n+1)\times m}$, to now also include sample $x_o$, a out-of-distribution example that contains variation along some nullspace feature $\mathbf{v}_o$. Looking at the response $\mathbf{z} = \mathbf{S}\mathbf{V}^T\boldsymbol{\theta}$, we note that $\mathbf{S}$ would now have an additional element, leading to a nonzero singular value along $\mathbf{v}_o$. However, evaluation of training dynamics in GS looks at optimizing $\boldsymbol{\theta}$ in the assumed linear approximation from equation 5 via minimizing ridge-regularized cross entropy loss, and any amount of ridge-regularization would make it such that the component of $\boldsymbol{\theta}$ along $\mathbf{v}_o$ would go down to exactly zero. As a result, the response $\mathbf{z} = \mathbf{S}\mathbf{V}^T\boldsymbol{\theta}$ along nullspace feature j would have nonzero corresponding values in $\mathbf{S}$ and $\mathbf{V}^T$, but a coefficient of 0 in $\boldsymbol{\theta}$. Therefore, this extension of the GS framework would predict a network response of zero along any nullspace component, at odds with the empirical results we find in this work.

## E    WEIGHT DECAY AND PROJECTION: AN ANALYSIS

Let us consider the weight decay regularization and how that relates to the eigenvalues of the weight matrices $W_l \in \mathbb{R}^{m \times K}$ learnt in different layers $l$. Specifically, weight decay minimizes the Frobenius norm $||W||_F$, which can also be written as $tr(W \cdot W^T)$. Consider the Singular Value Decomposition (SVD) of $W = U\Sigma V^T$, a rank $r$ for $W$, and let $u_i$ refer to a column of $U$. We can then write:

$$tr(W \cdot W^T) = tr(U\Sigma V^T V \Sigma U^T) \tag{9}$$

$$= tr(U\Sigma^2 U^T) \tag{10}$$

$$= tr(\sum_{i=1}^{r} \sigma_i^2 u_i \cdot u_i^T) \tag{11}$$

$$= \sum_{i=1}^{r} \sigma_i^2 tr(u_i \cdot u_i^T) \tag{12}$$

$$= \sum_{i=1}^{r} \sigma_i^2 \tag{13}$$

Where in the first line we used the fact that $V^T V = I$, then used the fact that $tr$ (trace) is a linear operation, and then used the fact that $||u_i|| = 1$.

Thus, adding weight decay to a network yields small singular values (in an L-2 sense). This, however does not yield sparsity. Thus, in some sense, "full-rank" information can potentially pass through multiple layers of a network with weight decay (this would not be possible if the rank $r$ was really small (or $\sigma_i = 0$) in contrast, for example.

This explains the complementary benefits of performing projection even in the presence of weight decay (as shown in the main paper).

## F    IN-DISTRIBUTION GENERALIZATION RESULTS

In this section, we report the performance of layer-oracle and oracle projection methods on in-distribution generalization in table table 8.

Interestingly, we find that the optimized basis consistently outperforms the other choices of bases. For example, on PACS, the optimized basis yields improvements of 0.9% as opposed to 0.3% for say, the PCA basis. Comparing the IID to the OOD setting, we find that the PCA basis yields much smaller improvements compared to the optimized basis. The ratio of the improvements in oracle from PCA basis to the optimized basis (averaged across all datasets) is 0.42 for IID compared to 0.79 for OOD. This suggests that, consistent with our intuition, the OOD datasets have more features occupying the nullspace in the PC basis and thus the PC basis performs better in OOD and worse in IID. However, considering the impact of the downstream function in addition to the variance (which our optimized basis does) can improve performance in IID setting as well.

## G   DATA-POISONING EXPERIMENT DETAILS

We perform our data poisoning experiments on the MNIST dataset. We split the MNIST dataset into two halves: a clean training set and a poisoned test set. We train a network with 4 convolutional layers, an average pooling layer, and 3 linear layers to perform well on the training set and poorly on the test set. All convolutional layers have a kernel size of $3 \times 3$, a padding of 1, and 64, 128, 128, and 128 channels respectively. The second convolutional layer uses a stride of 2, while all others use a stride of 1. Each convolution is followed by a ReLU nonlinearity, and a GroupNorm operation with 8 groups. After the convolutional layers, an average pooling layer with a kernel size of $1 \times 1$ and a stride of 1 is applied such that only a single value remains per channel. The resulting representation is fed to 3 linear layers of sizes $128 \times 64$, $64 \times 32$, and $32 \times 10$ respectively. Each linear layer save for the last is followed by a ReLU nonlinearity, and a batch normalization layer. To train the network, we use the Adam optimizer with $\beta_1 = 0.9$, and $\beta_2 = 0.999$. We train the network for 100 epochs with a batch size of 64 and learning rate of $1e-3$.

## H   HOW DO IMPROVEMENTS IN ACCURACY RELATE TO THE DISTANCE FROM THE SUBSPACE?

Do the observed improvements occur because we project to the subspace or is some other perturbation of the test data point $\mathbf{x}$ equally beneficial? To answer this, we consider an experiment where we partially project down the features $V'_m$. Formally speaking, each $\mathbf{v}^x$ can be written as a sum of two feature vectors $\mathbf{v}^x = \mathbf{v}_1^x + \mathbf{v}_2^x$ such that $\mathbf{v}_1^x$ lies in subspace $V_m$ and $\mathbf{v}_2^x$ lies in subspace $V'_m$. For $\lambda \in [0, 1]$, we consider the partial projection to $V_m$ as $\mathbf{v}_m^x = \mathbf{v}_1^x + \lambda \mathbf{v}_2^x$. If $\lambda = 1$, we recover the original feature vector (no projection) whereas $\lambda = 0$ yields the full projection to $V_m$ (methods discussed above).

fig. 7 shows the interpolation between $\lambda$ and the fraction of OOD improvement. It contains the plots for this experiment across the 12 best ERMs on PACS and TerraIncognita. Overall, we find that as we project out the component of feature vector in $V'_m$ and thus increase the distance from the subspace $V_m$, performance gradually decreases (roughly monotonically) for both the optimized basis as well

Table 8: Aggregate in-distribution generalization performance achieved by different bases on RotatedMNIST, PACS, TerraIncognita, and DomainNet. For each basis, we report the layer-oracle accuracy (**lo**) and the overall oracle performance (**o**).

|  | RotatedMNIST | PACS | TerraIncognita | DomainNet |
|---|---|---|---|---|
| ERM | 99.2 | 96.6 | 91.5 | 59.3 |

**ERM + Different Low-Rank bases $V_m$:**

|  | RotatedMNIST | | PACS | | TerraIncognita | | DomainNet | |
|---|---|---|---|---|---|---|---|---|
|  | **lo** | **o** | **lo** | **o** | **lo** | **o** | **lo** | **o** |
| Random | **99.2** | 99.2 | 96.7 | 96.8 | 91.6 | 91.7 | 59.4 | 59.4 |
| Jacobian | **99.2** | 99.3 | 96.6 | 97.0 | 91.9 | 92.2 | 59.6 | 60.3 |
| PCA | 99.1 | 99.3 | 96.8 | 97.0 | 91.8 | 92.2 | 59.7 | 60.4 |
| Low Rank $W_{l+1}$ | **99.2** | **99.4** | 96.7 | 96.9 | 91.5 | 91.6 | 59.7 | 60.4 |
| Optimized | 99.1 | **99.4** | **96.9** | **97.5** | **92.6** | **93.3** | **60.8** | **62.2** |

as the PCA basis. This suggests that improvements are likely due to the projection of the datapoints to the subspace $V_m$ as opposed to some other kind of structured perturbation.

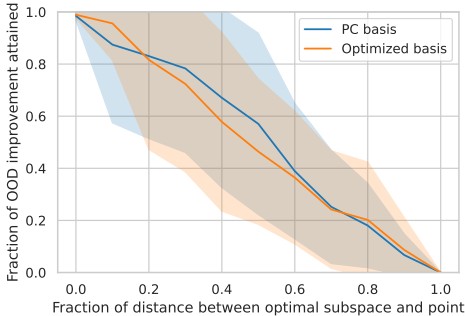 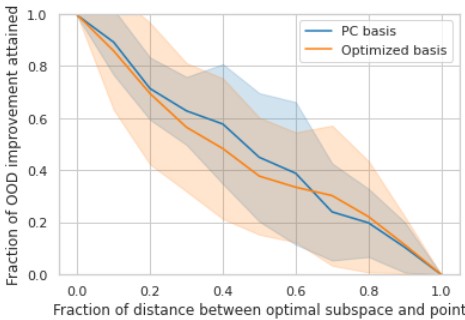

Figure 7: **Fraction improvement vs distance from optimal subspace:** For the optimal subspace $V_m$ (i.e. optimal layer and $m$), we project out $1 - \lambda$ fraction of the component orthogonal to $V_m$. For the resulting feature vector, we plot accuracy improvement divided by the optimal subspace accuracy improvement (y-axis) vs fraction of distance from the optimal subspace, $\lambda$ (x-axis). Solid lines show the mean fraction improvement over 12 DomainBed ERMs on PACS (left) and TerraIncognita (right), and shaded portions indicate the standard deviation.

## I    ARCHITECTURE FOR ROTATED-MNIST EXPERIMENTS

Following DomainBed, we use the ResNet-50 architecture for all experiments on PACS, TerraIncognita, and DomainNet, and a small CNN for experiments on Rotated-MNIST. The architecture for the small CNN consists of 4 convolutional layers, succeeded by global average pooling and one linear layer. The convolutional layers are of kernel size 3x3 and "same" padding, with 64, 128, 128, and 128 channels each. Each convolution layer is followed by a ReLU activation and a GroupNorm layer with 8 groups.

## J    ADDITIONAL PLOTS

In this section, we include plots of out-of-distribution accuracies after projection into all the different bases, for all networks, across Rotated-MNIST (fig. 8), PACS (fig. 9), TerraIncognita (fig. 10), and DomainBed (fig. 11). Following DomainBed methodology, we have three selected networks for each OOD domain (one from each independent trial), and plot the out-of-distribution accuracies after projection into different bases in the oracle layer of each network.

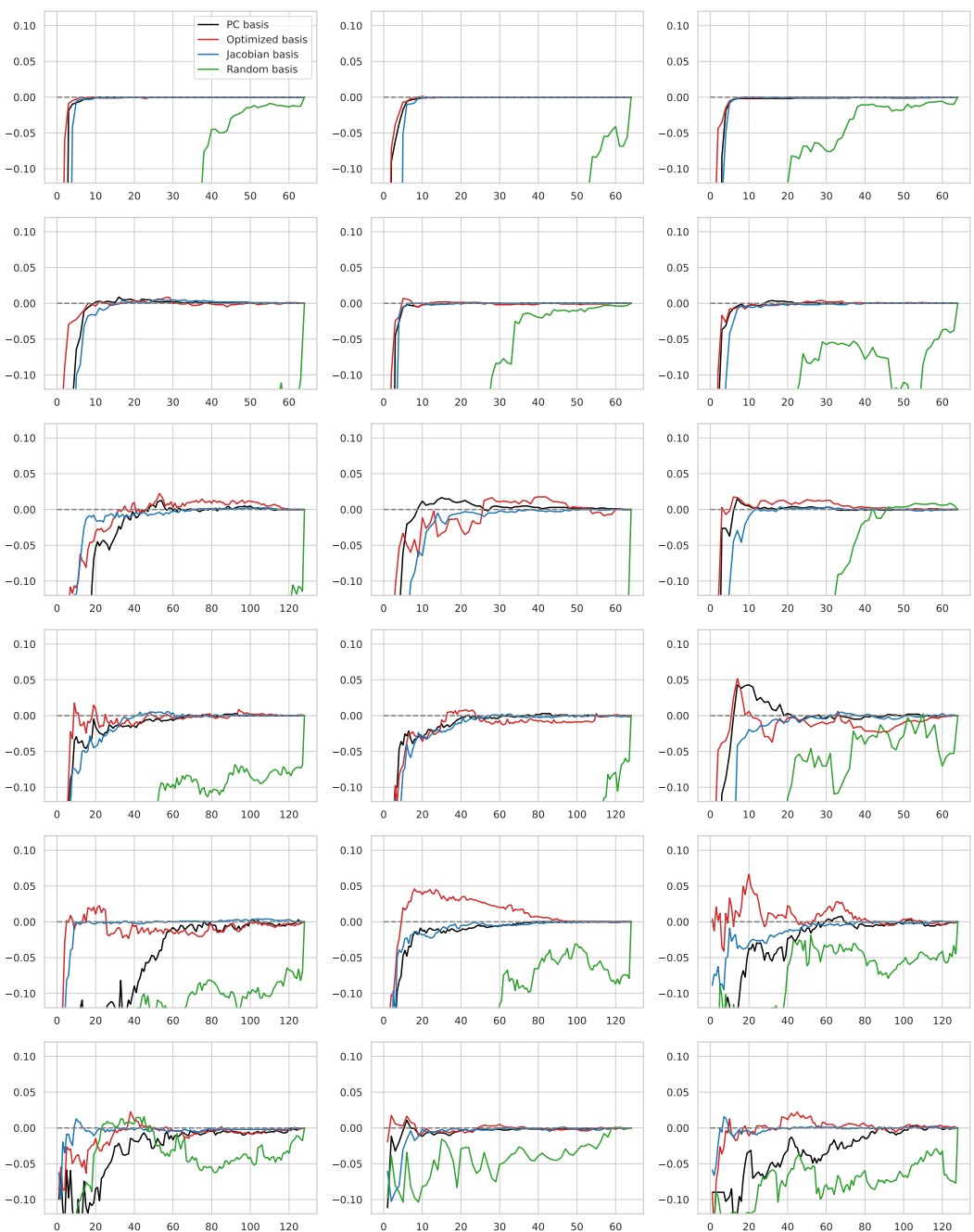

Figure 8: **Improvements in out-of-distribution classification relative to full network on Rotated-MNIST.** The x-axis is the rank of the subspace used for projection, while the y-axis is the improvement in OOD classification accuracy relative to the base network. Each row of plots corresponds to a different out-of-distribution test domain.

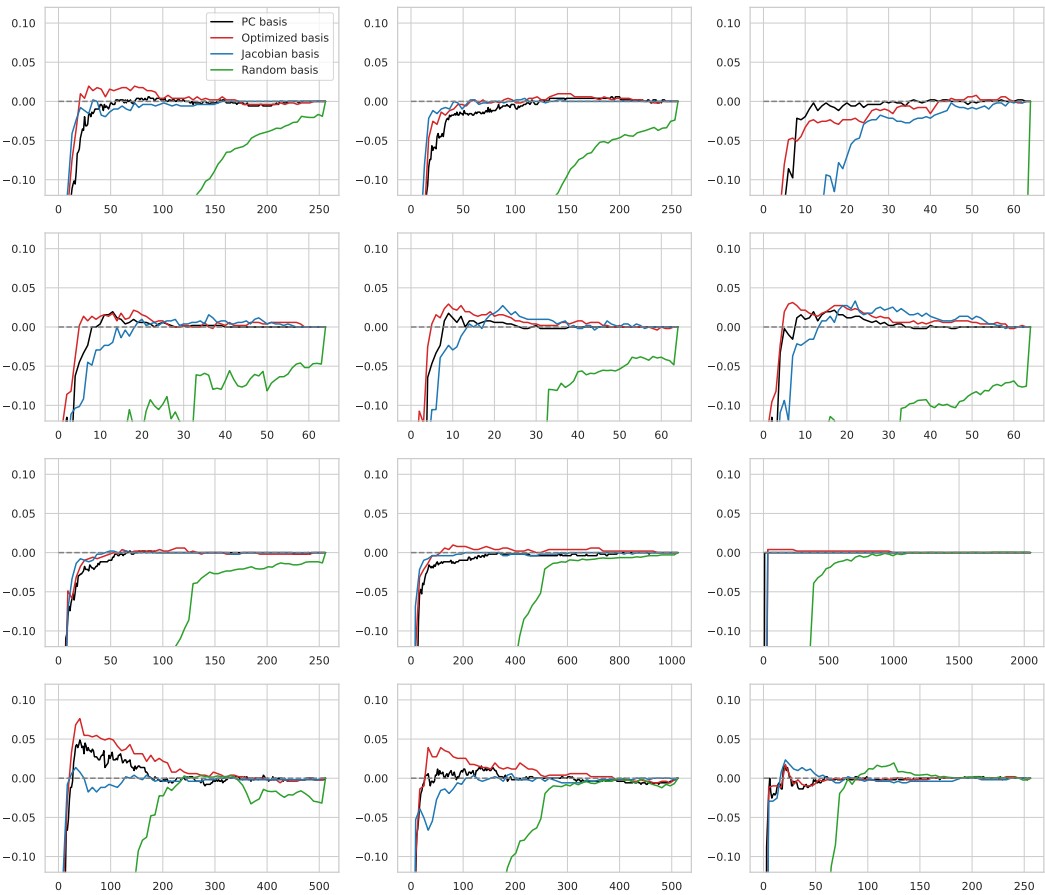

Figure 9: **Improvements in out-of-distribution classification relative to full network on PACS.** The x-axis is the rank of the subspace used for projection, while the y-axis is the improvement in OOD classification accuracy relative to the base network. Each row of plots corresponds to a different out-of-distribution test domain.

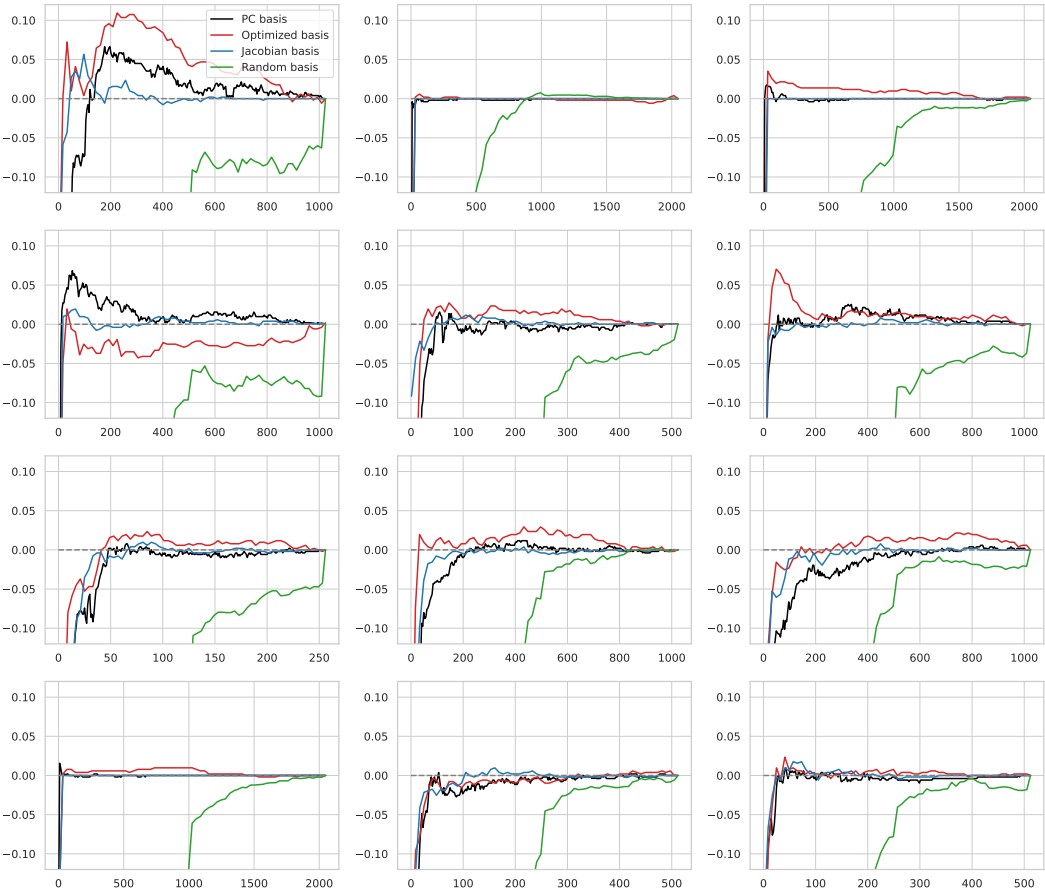

Figure 10: **Improvements in out-of-distribution classification relative to full network on TerraIncognita.** The x-axis is the rank of the subspace used for projection, while the y-axis is the improvement in OOD classification accuracy relative to the base network. Each row of plots corresponds to a different out-of-distribution test domain.

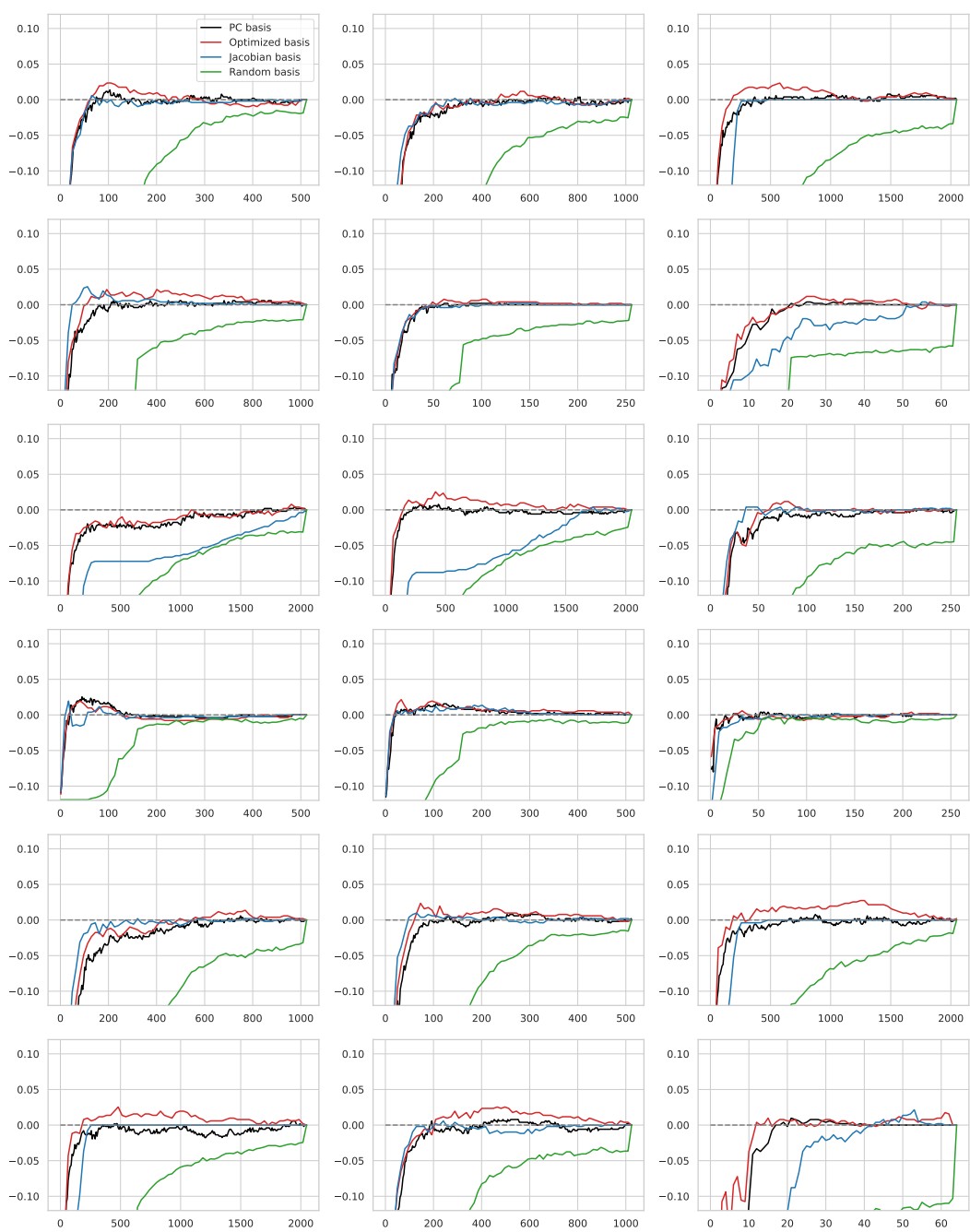

Figure 11: **Improvements in out-of-distribution classification relative to full network on Domain-Net.** The x-axis is the rank of the subspace used for projection, while the y-axis is the improvement in OOD classification accuracy relative to the base network. Each row of plots corresponds to a different out-of-distribution test domain.

## K    ROBUSTNESS TO ADVERSARIAL PERTURBATIONS

In the main paper, we showed how alleviating nullspace failure leads to improved OOD generalization. Can we also improve performance on adversarial examples generated to fool the full network by projecting to a limited number of components? To check this, we perform an experiment where we

evaluate oracle performance after projection on adversarial examples generated using AutoAttack (Croce & Hein, 2020), for a WideResNet-28-10 network trained on CIFAR-10. We consider attacks with a maximum $L_{inf}$ norm of $\epsilon = 8/255$. We report oracle accuracies for the PC and optimized bases, as well as corresponding CIFAR-10 test set performance in table 9, and plot accuracies for all choices of number of components in fig. 12. We additionally include in table 9 results for the augmentation driven adversarial defense of Gowal et al. (2021), a representative state-of-the-art method for this dataset and architecture. We find that both the PC basis and optimized basis recover a meaningful amount of the network's performance on the adversarial set.

|  | Test set | Adversarial set |
|---|---|---|
| ERM | 94.29% | 0% |
| PC basis | 78.52% | 46.48% |
| Optimized basis | 84.77% | 44.92% |
| Gowal et al. (2021) | 87% | 62% |

Table 9: Oracle performance after projection on the adversarial set, along with corresponding clean test set accuracies on CIFAR-10.

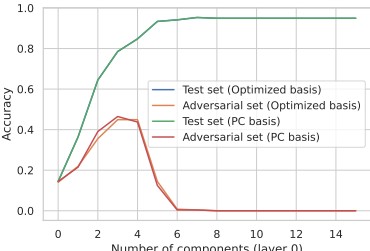

Figure 12: **Nullspace occupancy for adversarial examples.** Classification accuracy on CIFAR-10 test set and adversarial example set after projecting to the top-m components of the PC and optimized bases in layer 0, the oracle layer.

## L   PSEUDOCODE FOR COMPUTING ORACLE AND LAYER-ORACLE ACCURACIES

In this section we present pseudocode for the procedure used to report oracle and layer-oracle accuracies given the OoD and IID accuracies of a network for all choices of layer and number of components.

---

**Algorithm 1** Pseudocode for computing oracle and layer-oracle accuracies for one network given OoD and IID accuracies for all considered numbers of components and layers

---

1: **Input:** Layers $L = \{0, \ldots, l-1\}$, OoD accuracy matrix $\mathbf{A} \in \mathbb{R}^{l \times m}$, IID accuracy matrix $\mathbf{B} \in \mathbb{R}^{l \times m}$
2: Set oracle layer $l_o \leftarrow \arg\max_{i \in L} \max_{j \in \{0,\ldots,m-1\}} A_{ij}$
3: Set heuristic index $k \leftarrow \min_{j \in \{0,\ldots,m-1\}} j$ such that $\mathbf{B}[l_o, j] \geq 0.999 \times \mathbf{B}[l_o, m-1]$
4: Oracle accuracy $\mathbf{o} \leftarrow \max_{j \in \{0,\ldots,m-1\}} \mathbf{A}[l_o, j]$
5: Layer-oracle accuracy $\mathbf{lo} \leftarrow \mathbf{A}[l_o, k]$

---

## M   PSEUDOCODE FOR OPTIMIZING UNITARY MATRICES TO OBTAIN THE OPTIMIZED BASIS

In this section we present pseudocode for optimizing the objective in equation 3 using the unitary optimization procedure of Kiani et al. (2022).

---

**Algorithm 2** Pseudocode for obtaining the optimized basis

---

1: **Input:** Encoder $g$ to target layer, classifer $f$ from target layer, training mini-batches $\{(\mathbf{x}_i, \mathbf{y}_i)\}_{i=1}^N$, training loss $\mathcal{L}_{train}$, step size $\alpha$, PC basis $V_{pc}, \mu_{pc}$
2: Initialize basis $V \leftarrow V_{pc}, \mu \leftarrow \mu_{pc}$
3: **for** $i \in \{1, \ldots, N\}$ **do**
4:     Set loss $l \leftarrow 0$
5:     Get representation for target layer $\mathbf{z} \leftarrow g(\mathbf{x}_i)$
6:     **for** number of components $k \in \{1, \ldots, m\}$ **do**
7:         Project representation using eq 1 to get $\hat{\mathbf{z}}$, using $\mathbf{z}, V$, and $k$ components.
8:         Classify projected representation $\hat{y} \leftarrow f(\hat{z})$
9:         $l \leftarrow l + \mathcal{L}_{train}(\hat{y}, y)$
10:     **end for**
11:     Grad $\mathbf{g} \leftarrow \nabla_V l$
12:     Project gradient $\mathbf{p} \leftarrow 0.5 \times (\mathbf{g} - V\mathbf{g}^T V)$
13:     Update basis $V \leftarrow V \exp\left(-\alpha V^T \mathbf{p}\right)$
14: **end for**
15: Return $V, \mu$

---

