# OpenReview forum: "Don’t forget the nullspace! Nullspace occupancy as a mechanism for out of distribution failure"
_ICLR.cc/2023/Conference — ICLR 2023 poster_

### Official Review · Reviewer_k2wc · 2022-10-18

**Confidence:** 4
**Correctness:** 3
**Technical Novelty And Significance:** 3
**Empirical Novelty And Significance:** 3
**Recommendation:** 5

**Clarity, Quality, Novelty And Reproducibility:**

Clarity: The text is clearly written and easy to follow. The experimental design is clearly described.

Quality: Experiments are clearly described and executed. Results are discussed in the light of posed claims.

Novelty: The proposed approach to compression of representations of a previously trained models is novel to my knowledge.

Reproducibility: The main text contains little detail on how the problem posed in (3) is solved, which limits reproduction of results. I would suggest adding pseudo or actual code somewhere.

**Strength And Weaknesses:**

Pros:

+The paper reveals interesting issues in representation learning: representations preserve variance along directions that are irrelevant to the training risk.

+A simple and somewhat practical approach to compressing representations is proposed and, while authors focused on cases where training and testing distributions differ, it could also be useful in more standard i.i.d. cases.

+Under domain shifts, a large set of experiments shows consistent improvements over vanilla ERM, known to be a strong baseline for domain generalization benchmarks.


Cons:

-It is unclear to me how compressing representations would affect generalization under distribution shifts. To be clear, it seems to me that compressing away the risk null-space will affect generalization in the i.i.d. setting. As one would expect, focusing on the subset of risk minimizers that most compress data enables better generalization. However, why should we expect that to be helpful under distribution shifts? For instance, a widely discussed kind of shift known to affect performance is the existence of spurious features that correlate with labels only in the training sample (e.g., cow on grass vs. cow on the beach). Such spurious features would very likely be preserved after compression since they are so effective in minimizing the risk estimated with training data.

-Also related with the point made on the above, it's unclear which domain generalization setting is considered or which kinds of assumptions are expected to hold in order for the proposal to work. There are several domain generalization settings in the literature, each associated with a set of assumptions over the relationships between training and testing data sources, and different methods are better suited for different settings. It seems to me that the proposal focuses on the covariate shift scenario where data marginals shift, but not class conditional distributions. Is that so? And then again, why should we expect compressed representations to work better than their non-compressed counterparts in this scenario? It could be the case that compressing introduces domain invariance, but that should be verified. I would suggest measuring the H-divergence [1] or some other distance between train and test distributions with and without compression.

-It could be the case that the proposal, rather than offering some mechanism to enable learning from different domains, behaves as a regularizer in the standard sense: the projection step projects a model onto a "simpler" subset of risk minimizers, with lower-rank representations. If so, we should observe similar performance improvements on standard i.i.d. cases. Could it be the case that observed improvements over ERM are simply due to obtaining a better in-domain classifier?

-Minor comment: while in the text it is stated that the proposal does not require domain annotations, those are necessary to enable scheme such as leaving a training domain out for deciding on projection (hyper)parameters.

**Summary Of The Paper:**

In this work, authors hypothesize that a failure mode of neural networks corresponds to the case where models depend on features in the null-space of the training risk. That is, representations learned through ERM preserve non-negligible variance along directions that do not affect the training risk itself. Authors then claim that to be a source of generalization gap since train/test variations across said directions could affect the risk estimated on unseen data. To counter that, a scheme is proposed to learn extra parameters to compress representations and fine tune part of a network. Experiments are carried out under the domain generalization setting where improvements over vanilla ERM are observed.

**Summary Of The Review:**

The proposal is novel and interesting, and discusses properties of representations learned with very popular approaches. My main concerns lie in the fact that it's unclear how the proposal would induce better representations for domain generalization. While authors did report clear evidence of improving performance, it's unclear why that is.

---

> ### Author Response · Authors · 2022-11-19
> **Response 1/2**
>
> We thank the reviewer for thoughtful questions and suggestions. We clarify individual points below:
>
> > It is unclear to me how compressing representations would affect generalization under distribution shifts. To be clear, it seems to me that compressing away the risk null-space will affect generalization in the i.i.d. setting. As one would expect, focusing on the subset of risk minimizers that most compress data enables better generalization. However, why should we expect that to be helpful under distribution shifts? For instance, a widely discussed kind of shift known to affect performance is the existence of spurious features that correlate with labels only in the training sample (e.g., cow on grass vs. cow on the beach). Such spurious features would very likely be preserved after compression since they are so effective in minimizing the risk estimated with training data.
>
> Since nullspace components do not contribute to a reduction in training error, ERM does not differentiate between networks that handle nullspace components in different ways. Thus, ERM could result in learning networks whose output varies in arbitrary ways when observing features of OOD data with variation in nullspace components. For instance, in our toy example, the decision boundary of the network shifts with variation along nullspace component v_3. By projecting nullspace components out of representations of OOD data, we force networks to instead be invariant to variation along nullspace components. In contrast, components in the training subspace have impact on training error, and along these components the function behaves to minimize training error.
>
> We do not claim or expect that our method addresses all failure modes of generalization -- there are of course other failure modes of generalization that may exist within the training subspace, such as spurious correlation. We only show that since there is no meaningful signal for the network to learn along nullspace components, there is degradation in generalization performance when encountering OOD variation along nullspace directions.
>
>
> > Also related with the point made on the above, it's unclear which domain generalization setting is considered or which kinds of assumptions are expected to hold in order for the proposal to work. There are several domain generalization settings in the literature, each associated with a set of assumptions over the relationships between training and testing data sources, and different methods are better suited for different settings. It seems to me that the proposal focuses on the covariate shift scenario where data marginals shift, but not class conditional distributions. Is that so? And then again, why should we expect compressed representations to work better than their non-compressed counterparts in this scenario? It could be the case that compressing introduces domain invariance, but that should be verified. I would suggest measuring the H-divergence [1] or some other distance between train and test distributions with and without compression.
>
> We consider the domain generalization setting that is empirically observed in the image classification domain generalization benchmarks that we study, and show that the representations for domains not used for training have variance along nullspace components of the training subspace. It is not clear if different kinds of domain generalization settings and assumptions can be disambiguated in empirical settings, and discussion of which assumptions apply to these benchmarks is outside the scope of our work.
>
> We do not expect that representations are domain invariant after projection (there may very well be variance between domains within the training subspace). Furthermore, measures of domain invariance such as H-divergence are not empirically predictive of generalization for ERMs trained on DomainBed datasets [Vedantam et al. 2021].

---

> ### Author Response · Authors · 2022-11-19
> **Response 2/2**
>
>
> > It could be the case that the proposal, rather than offering some mechanism to enable learning from different domains, behaves as a regularizer in the standard sense: the projection step projects a model onto a "simpler" subset of risk minimizers, with lower-rank representations. If so, we should observe similar performance improvements on standard i.i.d. cases. Could it be the case that observed improvements over ERM are simply due to obtaining a better in-domain classifier?
>
> We do see an improvement over ERM in the IID case, although IID improvements are smaller than OOD ones, and there are some qualitative differences in the performance of different projection bases (appendix F). However, this does not imply that improvements in the OOD case are due to a better in-domain classifier. To improve OOD performance, we intervene on the training process, which is a common cause of both IID and OOD performance. This intervention leads to an improvement in both IID and OOD accuracy, which are known to be correlated [Miller et al. 2021].
>
> > Minor comment: while in the text it is stated that the proposal does not require domain annotations, those are necessary to enable scheme such as leaving a training domain out for deciding on projection (hyper)parameters.
>
> We agree that leave-one-out validation requires knowledge of domain labels, and have qualified our claim in the text to include this note.
>
> > Reproducibility: The main text contains little detail on how the problem posed in (3) is solved, which limits reproduction of results. I would suggest adding pseudo or actual code somewhere.
>
> Thanks for catching this. We include pseudocode for optimization of (3) in appendix X in the revised version.
>
>
> References:
>
> Accuracy on the line: on the strong correlation between out-of-distribution and in-distribution generalization. Miller et al. 2021. https://arxiv.org/abs/2107.04649
>
> An empirical investigation of domain generalization with empirical risk minimizers. Vedantam et al. 2021. https://proceedings.neurips.cc/paper/2021/file/ecf9902e0f61677c8de25ae60b654669-Paper.pdf

---

### Official Review · Reviewer_tTTa · 2022-10-24

**Confidence:** 4
**Clarity, Quality, Novelty And Reproducibility:** The overall presentation is clear and…
**Correctness:** 3
**Technical Novelty And Significance:** 3
**Empirical Novelty And Significance:** 3
**Recommendation:** 5

**Strength And Weaknesses:**

Strengths:

S1. The general problem of OOD detection is an important problem in reliable machine learning community.

S2. The main idea of identifying the existence of the failure mode across multiple networks and exploring different choices for characterizing the feature space provides an interesting insight to OOD detection.

Weaknesses:

W1. More discussion of the failure mode for models trained using different baselines and datasets are expected, besides models trained using ERM on DomainBed.

W2. How to quantify the particular failure mode of OoD generalization for discriminative classifiers.

W3. The authors are expected to analysize the relationship between using different backbone architectures, e.g., ViT, and identifying nullspace occupancy.


**Summary Of The Paper:**

This work identifies a particular failure mode of OoD generalization for discriminative classifiers and introduces the concept of nullspace occupancy when test data occupies the nullspace of low-rank feature space and captures the training set variability. This work finds that with a careful choice of the basis V_m, the out-of-distribution accuracy often improves by projecting out the remaining nullspace components for test data points.

**Summary Of The Review:**

This is a paper tackling an important problem. The observation is interesting and the writing is clear but these are some detailed issues that need to be clarified.

---

> ### Author Response · Authors · 2022-11-19
> **Response**
>
> We thank the reviewer for their suggestions. We address individual comments below:
>
> > W1. More discussion of the failure mode for models trained using different baselines and datasets are expected, besides models trained using ERM on DomainBed.
>
> DomainBed is the benchmark for OOD generalization studies, spanning many different datasets. State-of-the-art domain generalization methods are evaluated on DomainBed. For example, we include results for other baseline (representative of SOTA) methods including XYZ in table 2. In addition to our DomainBed results, we also demonstrate oracle improvements on ImageNet-R (section 4), data de-poisoning on MNIST (section 4), and improved oracle performance on adversarial robustness using CIFAR-10 (appendix K). This spans a large subset of datasets frequently used for evaluation in domain generalization studies. Regarding verifying nullspace failure on other baselines, note that ERM is already competitive with state-of-the-art domain generalization methods.
>
> > W3. The authors are expected to analysize the relationship between using different backbone architectures, e.g., ViT, and identifying nullspace occupancy.
>
> The focus of this paper is the relationship of nullspace occupancy with out-of-distribution, and so we focus on DomainBed methods, which use ResNet-50s and other small CNNs. We additionally have results on Wide Residual Networks on adversarial examples (Appendix K). We believe it is interesting future work to study whether this effect holds for ViT or other transformer based architectures on other domains such as language.
>
>
> > W2. How to quantify the particular failure mode of OoD generalization for discriminative classifiers.
>
> Quantifying the effect of the failure mode for discriminative classifiers in general is challenging, since the size of improvement after projection depends on the features learned by each network, and the training domains the network was trained on. For a given trained network and split of training domains and test domain, one way to quantify the failure mode would be to consider the difference in accuracy between the oracle projection and the ERM network without projection.

---

### Official Review · Reviewer_mtmw · 2022-10-25

**Confidence:** 3
**Correctness:** 4
**Technical Novelty And Significance:** 4
**Empirical Novelty And Significance:** 4
**Recommendation:** 8

**Clarity, Quality, Novelty And Reproducibility:**

Besides the key clarity issues I pointed out above, there are a few minor points:
- In the introduction, I am afraid I don't follow how "low rank simplicity bias" is bad for OoD generalization and how it relates to the two moons example.
- It seems unconventional to me to sandwich the related work section in between two sections that this paper's contributions.



**Strength And Weaknesses:**

Strengths
====
1. A concrete empirical and formal understanding of _why_ OOD failure occurs is important for developing principled OOD algorithms. The paper presents one such understanding.
2. The hypothesis of the null space failure mode is simple and intuitive to understand, and to the best of my knowledge not formalized in prior work.
3. The paper substantiates its claims with a wide variety of experiments on various OOD settings.
4. It's interesting that a mere projection of the representation (rather than say, reweighting as in [1,2]) results in accuracy improvements.
4. The paper provides excellent ablations/comparisons making the experiments technically sound. It provides relevant baselines for the projection technique, reports SoTA results for comparison, and also suggests and refutes alternative mechanisms by which the projection technique maybe working  (namely capacity control and l2 regularization).
5. The projection technique for identifying most important directions maybe of independent interest.

Weaknesses
===

1. The paper is currently missing key references/discussion connecting it to recent work that has proposed reweighting the top-layer features [ref. 1,2].

2. My other main complaint is regarding clarity. It's not clear to me how exactly the oracle accuracies are computed: are the bases computed on the target distribution? i.e., is L_train in (3) plugged to be the target distribution loss? I've assumed this to be the case, but I'm also confused if this is simply the original distribution's loss (which wouldn't make sense to me as an oracle).
- The motivation behind the oracle vs leave-one-out accuracies are presented was confusing and required multiple reads. I'd motivate the oracle accuracy as a theoretical upper bound, and the leave-one-out as a practically achievable value rather than something that "alleviates a limitation" of the oracle value.
- Perhaps a pseudo-code for the computation of oracle accuracy and layer-oracle accuracy may be helpful?

2. It's not clear why the ERM accuracies are different in Table 1 and 2. Perhaps I'm missing something about how ERM is run in leave-one-out settings.

3. I'd have also liked to see what happens to the in-distribution accuracy under these projection techniques. Are there plots for these?


[1]: Domain-Adjusted Regression or: ERM May Already Learn Features Sufficient for Out-of-Distribution Generalization, _Elan Rosenfeld, Pradeep Ravikumar, Andrej Risteski_ https://arxiv.org/abs/2202.06856
[2]: Last Layer Re-Training is Sufficient for Robustness to Spurious Correlations
_Polina Kirichenko, Pavel Izmailov, Andrew Gordon Wilson_ https://arxiv.org/abs/2204.02937


**Summary Of The Paper:**

===

This is to acknowledge that I've read the authors' clarifications to me and to the other reviewers. Thank you! I'd like to keep my score as it is; I've done my best to engage with the other reviewers to convince them as to why I think this paper deserves acceptance.

===

TL;DR: The paper hypothesizes a notion of OOD failure, and then presents an empirical technique to isolate it. The paper then demonstrates this failure mode in a real-world setting and presents preliminary ways to mitigate this failure.

More concretely:

1. The authors hypothesize the notion of "null space failure mode" where the presence of non-zero weights assigned to certain directions present in the training domain hurt OOD accuracy.

2. To test this in practice, given a hidden representation, they present a technique for finding a "basis" such that the top most directions are the most important for classification under a given distribution.

3. They then show in the DomainBed benchmark that as we project the OOD data only the top few basis vectors (learned using access to), there is surprisingly an _improvement_ in the OOD accuracy. Crucially, this improvement is comparable to SoTA values. This demonstrates the existence of null space failure.

4. To derive a practical algorithm, they propose a similar technique in the leave-one-out domain generalization setting. Here they report that the projection technique shows improvements of ERM (although not surpassing SoTA)



**Summary Of The Review:**

The paper presents an interesting formalization of OOD failure, an area where most work has been empirical and heuristical. The paper supports its claims via a variety of experiments with sufficient baselines/ablations. The core empirical result is interesting/surprising to me. I am hopeful that this will generate valuable follow-up work.

---

> ### Author Response · Authors · 2022-11-19
> **Response**
>
> We thank the reviewer for thoughtful questions and suggestions. We clarify individual points below:
>
>
> > It's interesting that a mere projection of the representation (rather than say, reweighting as in [1,2]) results in accuracy improvements.
>
> > The paper is currently missing key references/discussion connecting it to recent work that has proposed reweighting the top-layer features [ref. 1,2].
>
> We thank the reviewer for pointing out these references. We have now included discussion of connections between nullspace occupancy and feature reweighting methods to the related work section of the revision.
>
> > My other main complaint is regarding clarity. It's not clear to me how exactly the oracle accuracies are computed: are the bases computed on the target distribution? i.e., is L_train in (3) plugged to be the target distribution loss? I've assumed this to be the case, but I'm also confused if this is simply the original distribution's loss (which wouldn't make sense to me as an oracle). The motivation behind the oracle vs leave-one-out accuracies are presented was confusing and required multiple reads. I'd motivate the oracle accuracy as a theoretical upper bound, and the leave-one-out as a practically achievable value rather than something that "alleviates a limitation" of the oracle value. Perhaps a pseudo-code for the computation of oracle accuracy and layer-oracle accuracy may be helpful?
>
> We thank the reviewer for bringing up this concern. To improve clarity, we have updated the descriptions of oracle and leave-one-out accuracies in the experimental setup section, and added pseudocode for computing the oracle and layer-oracle accuracies in Appendix L.
>
> Both the oracle and layer-oracle numbers use the same optimized basis computed once for each network on the original distribution’s loss -- the “oracle” and “layer-oracle” terminology corresponds to picking a choice of layer and number of components to perform projection over to achieve maximum out-of-distribution accuracy. The “oracle” assumes access to out-of-distribution accuracy when selecting a projection layer and number of components.
>
> For computing bases, we use the original (training) distribution’s loss. The goal is to identify features in the training domain’s nullspace, which requires computing the basis over the training domain. We now include pseudocode for how to optimize equation (3) in Appendix M.
>
> > It's not clear why the ERM accuracies are different in Table 1 and 2. Perhaps I'm missing something about how ERM is run in leave-one-out settings.
>
> The difference in ERM accuracies between the two tables for PACS and TerraIncognita is due to using fewer training domains when training ERMs for leave-one-out evaluation. In the oracle/layer-oracle experiments, for each selected test domain, we train ERM networks on all remaining domains and test on the selected test domain. In contrast, in leave-one-out experiments, for each selected test domain, we reserve an additional domain for leave-one-out selection, and train on one fewer training domain. For Rotated-MNIST, where all networks are trained on 0-degree data, we hold out the leave-one-out domain from the test-set evaluation, leading to a different average performance for ERM.
>
> > I'd have also liked to see what happens to the in-distribution accuracy under these projection techniques. Are there plots for these?
>
> We discuss oracle results for in-distribution accuracies for DomainBed datasets in Appendix F. We found that projection improves performance in the in-distribution setting as well. We found that the gains in accuracy for IID data are typically smaller than those in the OOD setting. Additionally, we found that when comparing the IID to the OOD setting, the PCA basis yields much smaller improvements compared to the optimized basis.
>
> > In the introduction, I am afraid I don't follow how "low rank simplicity bias" is bad for OoD generalization and how it relates to the two moons example.
>
> For the low-rank simplicity bias, what we mean is that the feature spaces learnt by neural networks (in some latent space) are similar to the two-moons example (in the sense that both are low rank). In such a case, the downstream network function (decision boundaries shown in Fig. 1) has potential to generalize poorly in the nullspace.
>
> > It seems unconventional to me to sandwich the related work section in between two sections that this paper's contributions.
>
> Thanks for catching this -- we have moved the related work section to after the results section.

---

### Official Review · Reviewer_L6zm · 2022-10-28

**Confidence:** 3
**Correctness:** 3
**Technical Novelty And Significance:** 3
**Empirical Novelty And Significance:** 3
**Recommendation:** 5

**Clarity, Quality, Novelty And Reproducibility:**

The writing was clear and the paper was overall enjoyable to read. The main observation and method is novel, to my knowledge.

**Strength And Weaknesses:**

Figure 1 is not very intuitively clear to me. I get that because the model was trained on data with v_3=1, its predictions on v_3=10 did not matter, and v_3 is an abstraction of directions of unseen variation in general. However, I’m not sure showing two classifiers side by side is the best way to convey this idea.

Experiments show improvements of about 1~2% OOD accuracy, which outperforms three existing approaches for OoD generalization. The experiment on data poisoning is quite compelling.

Question for authors:
(1) Do you have any hypotheses that would resolve the difference with the gradient starvation phenomenon? Using the notation in page 9, why are features like v_2 ignored while v_3 is attended to (in a suboptimal way, of course)?
(2) Looking at figs 9-12, many layer/test domain combinations seem to not improve after projection. Do you have a sense for why only some layers benefit from projection, and why the location of those layers is quite different for each test domain?

Minor comments
- Page 2: your wrapping carried over the the next page, and this page starts with awkward indentation.

**Summary Of The Paper:**

This paper makes the observation that some models fail to OoD generalize because discriminative classifiers for OoD test data lie on the null space of learned features. They propose a simple method to avoid this failure: project features onto a low-rank subspace that reflects what was seen in the source data.

**Summary Of The Review:**

I think the main observation is interesting and the proposed method makes sense in the context of the observation. I had some conceptual questions about the observation, and would be happy to increase my score if those are addressed.

---

> ### Author Response · Authors · 2022-11-19
> **Response**
>
> We thank the reviewer for thoughtful questions and suggestions. We clarify individual points below:
>
> > Do you have any hypotheses that would resolve the difference with the gradient starvation phenomenon? Using the notation in page 9, why are features like v_2 ignored while v_3 is attended to (in a suboptimal way, of course)?
>
> Intuitively, we think the difference comes from whether the features v_2 can reduce training error (which they can in gradient starvation but are ignored), v.s. whether the features like v_3 have no influence over the training loss (in which case the problem we are solving is underconstrained).
>
> More formally, we expanded on this difference in Appendix D. We found that the gradient starvation theoretical framework cannot be easily extended to features outside of the training subspace.
>
> The set of features considered in the GS framework are obtained via singular value decomposition of the neural tangent random feature matrix, defined over examples from the training set. These features do not contain features outside of the training subspace (such as v_3), and the GS framework does not necessarily prescribe if v_3 will be ignored or attended to.
>
> If we attempt to extend the GS framework to apply to nullspace features by instead computing the NTRF over training examples and one additional OOD example (that creates variation in the v_3 feature), then the corresponding nullspace feature in the NTRF feature set would be ignored because of the ridge regularization in the GS loss function (ridge-regularized cross entropy loss).
>
> Note also that features in GS and in nullspace occupancy are in different spaces, and there may not be a direct application of GS theory to the bases considered in our work in the first place.
>
>
> > Looking at figs 9-12, many layer/test domain combinations seem to not improve after projection. Do you have a sense for why only some layers benefit from projection, and why the location of those layers is quite different for each test domain?
>
> Our findings are that projection helps in most networks for at least one layer. A priori, without the nullspace occupancy being an issue, there is no reason why the projection should help in the first place. Thus, our focus was on exposing the nullspace occupancy as a potential mechanism for OOD generalization.
>
> Why do some layers only benefit from projection? One hypothesis we have for why only some layers benefit from projection is that once you’re already “in the nullspace” in some layer j, applying projection in some layer i > j may not be helpful. Additionally, projection may sometimes not yield a large improvement in early layers of the network since the corresponding feature spaces are in a sense only slightly different from the input image space, and it may be the case that nullspace features at a higher level of abstraction are more helpful to eliminate.
>
> Why do differences exist between test-domains? Each test-domain corresponds to a different set of training domains. In general, we do not prescribe the layer where nullspace features might be observed -- the oracle layers could be quite different between any 2 networks. However, within a single set of training domains, there are sometimes similar patterns in where nullspace features are observed. Speculatively, this may be due to different networks representing input features at similar levels of abstraction in each layer, leading to similarities in where nullspace features are observed.
>
>
> > Figure 1 is not very intuitively clear to me. I get that because the model was trained on data with v_3=1, its predictions on v_3=10 did not matter, and v_3 is an abstraction of directions of unseen variation in general. However, I’m not sure showing two classifiers side by side is the best way to convey this idea.
>
> Thanks for catching this. We have updated the caption for figure 1 in the revised version to clarify that the two panels are the same classifier, evaluated on different values of v_3.
>
>
> > Page 2: your wrapping carried over the the next page, and this page starts with awkward indentation.
>
> Thanks -- we have fixed this in the revised version.

---

### Decision · Program_Chairs · 2023-01-20

**Decision:**

Accept: poster

**Justification For Why Not Higher Score:**

Unclear whether the gains are significant enough to improve state-of-the-art. Generally, whether this is high impact work is still unclear.

**Justification For Why Not Lower Score:**

N/A

**Metareview: Summary, Strengths And Weaknesses:**

The paper uncovers a novel mechanism contributing to poor out of distribution performance in discriminative classifiers, which is related to the nullspace of the model. The mechanism is demonstrated across datasets, including RotatedMNIST, PACS, TerraIncognita, DomainNet, and ImageNet-R. The experiments are carefully designed and help corroborate the proposed mechanism.

The reviewers agree that the identified mechanism is novel and interesting, and that the experiments are thoughtfully designed to corroborate it and support its causal impact on out of distribution performance.

Reviewers raised several concerns, but in my opinion Authors' have addressed them in the rebuttal. In particular, there was concern whether the out of distribution gains could be explained simply by the improvement in iid performance. This should be investigated more carefully, but results in the paper suggest that the answer is no. Most convincingly, the results are significantly impacted by the choice of the projection method.

One weakness of the paper is that the method was not investigated in combination with state-of-the-art methods. Hence, it is unclear whether the approach allows to improve the current state-of-the-art.

All in all, it is my pleasure to recommend acceptance of the paper. Thank you for your submission, and please remember to address the reviewers' feedback in the camera ready version.

**Note From Pc:**

if the above contains the word "oral" or "spotlight" please see: "oral" presentation means -> notable-top-5% and "spotlight" means -> notable-top-25%. As stated in our emails, we are disassociating presentation type from AC recommendations